# Online Knapsack with Frequency Predictions

**Sungjin Im**
Electrical Engineering and Computer Science
University of California, Merced
sim3@ucmerced.edu

**Ravi Kumar**
Google Research
Mountain View, CA
ravi.k53@gmail.com

**Mahshid Montazer Qaem**
Electrical Engineering and Computer Science
University of California, Merced
mmontazerqaem@ucmerced.edu

**Manish Purohit**
Google Research
Mountain View, CA
mpurohit@google.com

## Abstract

There has been recent interest in using machine-learned predictions to improve the worst-case guarantees of online algorithms. In this paper we continue this line of work by studying the online knapsack problem, but with very weak predictions: in the form of knowing an upper and lower bound for the number of items of each value. We systematically derive online algorithms that attain the best possible competitive ratio for any fixed prediction; we also extend the results to more general settings such as generalized one-way trading and two-stage online knapsack. Our work shows that even seemingly weak predictions can be utilized effectively to provably improve the performance of online algorithms.

## 1 Introduction

An algorithm designer's ultimate goal is to develop algorithms that are reliable and work well in practice. High reliability has been the major focus of discrete algorithmic research that primarily seeks to give strong worst-case guarantees for *all* inputs. Unfortunately, such algorithms treat all input instances on an equal footing—even pathological ones that rarely occur in practice—and do not necessarily exploit the latent structure that can be present in realistic instances. In contrast, machine learning has demonstrated amazing successes in many real-world applications, yet occasionally exhibits unacceptable failures on specific (worst-case) instances.

Learning-augmented algorithm design has recently emerged as a model to achieve both reliability and worst-case guarantees, particularly in the online setting. In this model, an online algorithm is given certain predictions on the input it will face in the future, and the algorithm then makes irrevocable decisions as the actual input is revealed one by one. Ideally, the algorithm should perform close to the optimum when the predictions are good, and yet possess worst-case guarantees even when the predictions are malicious. Recent work has shown that this is achievable for online problems such as caching [17, 26, 29], ski-rental [2, 14, 22], scheduling [6, 15, 22, 28], load balancing [23], secretary problem [5], metrical task systems [4], set cover [7], flow/matching [24], bin packing [3], etc.

Most prior work on learning-augmented online algorithms assumes that we are given as prediction a specific value of key parameters of the online instance, e.g., the number of days to ski in the ski-rental problem or the next request time of a page in the caching problem. In this paper, we take a relaxed approach and assume that the prediction only gives a *range* of values for key parameters. This is a natural way to model weak predictions where the predictor has high uncertainty and can only provide a reliable range in which the value must lie. Given such weak predictions, we seek to obtain beyond worst-case guarantees for all inputs *respecting* the prediction, by possibly exploiting the structure

35th Conference on Neural Information Processing Systems (NeurIPS 2021).

provided by these predictions. In fact, we seek algorithms that give worst-case guarantees for all inputs while promising sharper bounds for instances that indeed respect the predictions. In this paper, we design such learning-augmented algorithms with weak predictions for the online knapsack problem, which is a basic problem of practical importance [31, 30].

**Knapsack Problem and the Prediction Model.** In the classic *online knapsack problem*, we have a knapsack of unit capacity[i] and $n$ items that arrive online. Each item $i$ has a *profit* $p_i \geq 0$ and a *size* $0 \leq s_i \leq 1$. For convenience, we define the *value* of an item, $v_i := p_i/s_i$, to be the profit of the item per unit size. We assume that the value of each item lies in a range known a priori, i.e., we have $v_{\min} \leq v_i \leq v_{\max}$ for some known constants $v_{\min}$ and $v_{\max}$.[ii] When an item arrives, an online algorithm must make an irrevocable choice of whether to accept or reject the item. The goal is to select a subset of items of the highest total profit whose total size is at most the knapsack capacity.

Recall that an online algorithm is said to have a *competitive ratio c*, or equivalently, be *c-competitive* if the algorithm's objective (profit) is at least $c$ times the total profit obtained by the optimal solution for all inputs. Zhou, Chakrabarty, and Lukose [31] gave a $1/(1 + \ln \frac{v_{\max}}{v_{\min}})$-competitive algorithm (that we call *ZCL*) under the assumption that all items have infinitesimal sizes (or can be accepted fractionally) and further showed that no algorithm can obtain a better competitive ratio.

Online knapsack is a fundamental problem in online algorithm design and finds applications in many real-world settings, e.g., in the context of online auctions [9, 31] and charging electric vehicles [30]. For these practical applications however, the worst-case guarantees offered by standard competitive analysis are too pessimistic to be useful. Indeed, practical instances are not typically worst-case and further one often possesses some foreknowledge of the kind of items that are expected to arrive in the future. In an online ad-auction, for instance, the advertiser can reliably predict a range such that the number of queries in a day for a particular keyword lies within the range. In this work, we explore formal models to utilize such predictions to obtain provably good algorithms for online knapsack beyond what is possible via classical competitive analysis. In particular, we introduce and study the online knapsack problem with frequency predictions (KNAPSACK-FP), described next.

Let $V$ denote the set of all possible values that items may take, and $v_{\min} = \min(V)$ and $v_{\max} = \max(V)$ denote minimum and maximum values respectively. We may assume that the items have discrete values by rounding them to the closest power of $(1 + \epsilon)$ for some $\epsilon > 0$, yielding $|V| = O(\log(v_{\max}/v_{\min}))$ and making the competitive ratio only marginally worse. For any value $v \in V$, let $s_v = \sum_{i | v_i = v} s_i$ denote the total size of items with value $v$. In the *online knapsack with frequency predictions* problem, for each value $v \in V$, the *frequency predictions* $P = \{\ell_v; u_v\}_{v \in V}$ provided to the algorithm are a lower bound $\ell_v$ and an upper bound $u_v$ such that $\ell_v \leq s_v \leq u_v$. We say that the algorithm has a competitive ratio of $\alpha$ for KNAPSACK-FP with prediction $P$ if the algorithm's profit is at least $\alpha$ times the optimum for all inputs respecting the given frequency prediction $P$. Our goal is to design an online algorithm with the best competitive ratio for this prediction model.

## 1.1 Our Results and Techniques

Our main result is a nearly optimal algorithm for KNAPSACK-FP.

**Theorem 1.** *For* KNAPSACK-FP*, given any frequency predictions, we can find an algorithm of competitive ratio* $\alpha^* - \frac{v_{\max}}{v_{\min}} \cdot (|V| + 1) \cdot \max_i s_i$*, where* $\alpha^*$ *is the best possible competitive ratio for the problem (with predictions).*

Our algorithm, called SENTINEL, pre-computes a *budget* $b_v$ for each value $v$ and accepts items that maintain the invariant that the total size of accepted items of value at most $v$ is at most $B_v := \sum_{v_{\min} \leq x \leq v} b_x$ for all $v \in V$. In some sense, $B_v$ can be viewed as a *sentinel* to guard against accepting too many low-valued items. The key intuition underlying our algorithm is that the knowledge regarding the arrival of future higher-valued items enables it to not over-allocate capacity to earlier lower-valued items. Our algorithm can also be implemented using only $O(\log |V|)$ time per item. We note that the additive loss of $\frac{v_{\max}}{v_{\min}} \cdot (|V| + 1) \cdot \max_i s_i$ in the competitive ratio in Theorem 1 is

---

[i]This is without loss of generality by scaling all sizes.

[ii]Such an assumption is necessary since otherwise we cannot hope for a bounded competitive ratio in the adversarial model [27, 31].

due to a reduction to a continuous version of the problem (as discussed in Section 2) and this quantity approaches zero as $\max_i s_i \to 0$.

Interestingly our algorithm has a distinguishing feature from the ZCL [31] algorithm for the standard online knapsack. The ZCL algorithm constructs a *threshold* function $\Psi(t) = (e \cdot v_{\max}/v_{\min})^t \cdot (v_{\min}/e)$ a priori and accepts an item online if its value is no smaller than $\Psi(t)$ when $t$ is the knapsack capacity that has been consumed. In contrast, we show that *no* such algorithm that maintains a threshold function based on the capacity used can give the optimum algorithm (Section 3.2). We note that by combining our algorithm and ZCL (say by using a $\gamma$ fraction of the capacity to run our algorithm and the other $1 - \gamma$ fraction to run ZCL), we can simultaneously obtain a worst-case guarantee of $\frac{1-\gamma}{1+\log(\frac{v_{\max}}{v_{\min}})}$ as well as our improved guarantees for instances that respect the prediction. Further, we can easily extend our result to show that our algorithm has a small loss in the competitive ratio when the predictions are slightly incorrect, for example, when we have $\ell_v/(1+\epsilon) \le s_v \le u_v(1+\epsilon)$.

**Extensions.** We extend our model and results to two generalizations of the knapsack problem: the *generalized one-way trading* problem and the *two-stage online knapsack* problem. In generalized one-way trading [13, 30], each item of type $t$ is associated with a certain concave function $f_t$ with $f_t(0) = 0$. If an algorithm accepts $s$ amount of items having type $t$, then it obtains profit $f_t(s)$. The goal, as always, is to accept a set of items subject to the knapsack capacity to maximize the total profit. This is a well-studied problem, motivated by converting a given amount of money from one currency to another, in the presence of time-varying conversion rates.

The two-stage online knapsack is a generalization that we introduce in this paper. The key difference from the standard setting is that after seeing all the items, the online algorithm is given another chance in a second stage to accept any items that it rejected before. But, an item accepted in the second stage gives only a $\lambda \le 1$ fraction of its original profit. The two-stage knapsack effectively interpolates between the online and offline problems: it becomes the standard online knapsack problem if $\lambda = 0$, and the offline problem if $\lambda = 1$. The two-stage setting has natural applications. For example, while clients typically prefer to know if their requests will be accepted instantly, some might choose to wait further for a discount, albeit with the possibility of getting no service if the service provider accepts other competing requests arriving later.

For both extensions we generalize our results and obtain the best competitive algorithms given frequency predictions (Sections 4 and 5). This shows that our prediction model and analysis could find more applications, providing beyond-worst-case guarantees even with weak predictions.

**Experiments.** In Section 6, we conduct experiments on synthetic data sets, and show our algorithm augmented with frequency predictions considerably outperforms the classic worst-case ZCL algorithm.

## 1.2 Other Related Work

The knapsack problem has been extensively studied in the literature. For a comprehensive survey of the results on the offline knapsack problem, see [18]. The offline version of the standard knapsack problem and most of its variants are well-known to be NP-hard.

The classic online knapsack problem falls into a general class of problems called online packing problems and one can obtain an $\Omega(1/\ln(v_{\max}/v_{\min}))$-competitive algorithm via the online primal-dual technique [10, 11]. For a nice survey of online primal-dual technique and its applications for various packing and covering problems, see [12]. Sun et al. [30] generalize the threshold algorithm in [31] by introducing a control parameter that adjusts how fast the threshold grows. They empirically learn the best parameter within a certain range to retain some worst-case guarantees, but do not provide optimality. Kong et al. [21] empirically show that reinforcement learning can find a good threshold for the online knapsack.

Another line of work considers the stochastic online knapsack problem where items' profits and sizes are drawn from a distribution (e.g., see [27, 25, 20] and their follow-up works). In the random arrival model, the items are assumed to be chosen adversarially but arrive in a uniformly random order. In this model, one can obtain constant competitive algorithms [1, 19] and further if all items have an infinitesimal size, one can obtain nearly-optimal online algorithms [19]. There are many other

variants studied and we only mention a few here, e.g., removable online knapsack problems [16] and online knapsack with reservation costs [8].

## 2 Preliminaries

To ease the presentation, we will consider the following relaxed model, which differs from KNAPSACK-FP in two aspects: first, we allow an item to be accepted fractionally and second, we allow an item to have any value in $[v_{\min}, v_{\max}]$, not from a discrete set. The second relaxation is purely for ease of presentation and it preserves the competitive ratio. The first relaxation results in a small additive loss in the competitive ratio since at most one item is accepted fractionally among those of the same value and all items are assumed to be small. We now formalize the relaxed model.

**Continuous-Valued Online Knapsack with Frequency Predictions (C-KNAPSACK-FP).** In the continuous version, we assume that there is a continuum of items, each having *infinitesimal* size and some value $v \in [v_{\min}, v_{\max}]$. In a particular problem instance, for any value $v \in [v_{\min}, v_{\max}]$, let $s(v)\mathrm{d}v$ be the total size of items having value in the range $[v, v + \mathrm{d}v]$ for some sufficiently small $\mathrm{d}v$. For each value $v \in [v_{\min}, v_{\max}]$, we are given a lower bound[iii] $\ell(v)$ and an upper bound $u(v)$ such that $s(v) \in [\ell(v), u(v)]$. We assume that $s(v), \ell(v), u(v)$ are all continuous in $v$. Define $S(v) := \int_{x=v_{\min}}^{v} s(x)\mathrm{d}x$, $L(v) := \int_{x=v_{\min}}^{v} \ell(x)\mathrm{d}x$, and $U(v) := \int_{x=v_{\min}}^{v} u(x)\mathrm{d}x$, which are differentiable at all $v \in [v_{\min}, v_{\max}]$ by definition. As in the standard model, we have a knapsack of unit capacity, and the goal is to select a subset of items of total size at most one that maximizes total value. The following relates the competitive ratio of the standard and the continuous-valued versions.

**Lemma 2.** *Given a prediction $P = \{\ell_v; u_v\}_{v \in V}$ for KNAPSACK-FP, we can create a prediction $P' = \{\ell(v); u(v)\}_{v \in [v_{\min}, v_{\max}]}$ for C-KNAPSACK-FP in polynomial time such that*

1. *An $\alpha$-competitive algorithm for C-KNAPSACK-FP with prediction $P'$ implies an $(\alpha - \frac{v_{\max}}{v_{\min}} \cdot |V| \cdot \max_i s_i)$-competitive algorithm for KNAPSACK-FP with prediction $P$.*
2. *If no algorithm has a competitive ratio better than $\alpha$ for C-KNAPSACK-FP with prediction $P'$, then no algorithm has a competitive ratio better than $\alpha + \max_i s_i$ for KNAPSACK-FP with prediction $P$.*

Thus, if we find an algorithm with the optimum competitive ratio for C-KNAPSACK-FP, then we will have Theorem 1. In the rest of the paper we only consider the continuous version for convenience.

## 3 The SENTINEL algorithm

Let $\{\ell(v); u(v)\}_{v \in [v_{\min}, v_{\max}]}$ be a fixed set of predictions; this defines a family of knapsack instances, where each instance is a sequence of items, each with a profit and size. Let $\alpha$ denote the desired competitive ratio of our algorithm. We first derive an online algorithm that guarantees a competitive ratio of $\alpha$ assuming one exists and then discuss how one can find the optimal such competitive ratio $\alpha^*$. For any instance $\mathcal{I}$, let $\mathrm{OPT}(\mathcal{I})$ denote the total value obtained by the optimal solution for instance $\mathcal{I}$. To systematically derive our algorithm, it is instructive to consider the following family of instances. For any value $v \in [v_{\min}, v_{\max}]$, let $\mathcal{M}(v)$ denote an instance containing the maximum possible amount of items of value at most $v$ and the minimum possible amount of items of value larger than $v$. Formally, $\mathcal{M}(v)$ is an instance such that $s(x) = u(x)$ for all $x \le v$ and $s(x) = \ell(x)$ for all $x > v$. Further, items in $\mathcal{M}(v)$ arrive in the following order: first $\ell(x)$ items of value $x \in [v_{\min}, v_{\max}]$ arrive in non-decreasing order of value, followed by $u(x) - \ell(x)$ items of value $x \le v$ arriving in non-decreasing order. Note that we consider $\mathcal{M}(v)$ only to derive our algorithm.

**Step 1.** Let $v^*$ denote the largest value such that

$$\int_{x=v^*}^{v_{\max}} x \cdot \ell(x)\mathrm{d}x = \alpha \cdot \mathrm{OPT}(\mathcal{M}(v^*)). \tag{1}$$

---

[iii]We use the functional notation (e.g., $\ell(v)$) for the continuous case and the subscript notation (e.g., $\ell_v$) for the discrete case.

Intuitively $v^*$ denotes the largest value such that an algorithm is guaranteed to be $\alpha$-competitive even if it does not accept any items of value less than $v^*$. Note that since we assume that the functions $\ell(\cdot)$ and $u(\cdot)$ are continuous, such a value $v^*$ is guaranteed to exist by the intermediate value theorem.

**Step 2.** Consider an adversary that has constructed an instance of the form $\mathcal{M}(x)$ for some unknown value $x$. By instead constructing an instance of the form $\mathcal{M}(x + \mathrm{d}x)$, the adversary can gain an additional profit of $\frac{\mathrm{d}}{\mathrm{d}x}\mathrm{OPT}(\mathcal{M}(x))$. In order to maintain a competitive ratio of $\alpha$, our algorithm must accept enough additional items of value $x$ so that it receives a profit of at least $\alpha \cdot \frac{\mathrm{d}}{\mathrm{d}x}\mathrm{OPT}(\mathcal{M}(x))$. Motivated by this discussion, we define a function $\tau : [v^*, v_{\max}] \to [0, \infty)$ to determine the additional amount of items of a particular value that our algorithm accepts.

$$\tau(x) = \frac{\alpha}{x}\frac{\mathrm{d}}{\mathrm{d}x}\mathrm{OPT}(\mathcal{M}(x)), \qquad \forall x \in [v^*, v_{\max}]. \tag{2}$$

**Step 3.** For any value $x \in [v_{\min}, v_{\max}]$, we define a budget function as follows.

$$b(x) = \begin{cases} 0, & \forall x < v^* \\ \ell(x) + \tau(x), & \forall x \geq v^* \end{cases}; \text{ and } B(v) = \int_{x=v_{\min}}^{v} b(x)\mathrm{d}x \quad \forall v \in [v_{\min}, v_{\max}]. \tag{3}$$

For any value $v$, the budget function $B(v)$ defined above prescribes the maximum total amount of items of value at most $v$ that must be accepted. Formally, let $A(v)$ denote the total amount of items of value at most $v$ that has been accepted by the online algorithm so far. Then, when an item of value $v$ arrives, the algorithm accepts the item if and only if $A(x) < B(x)$ for all $x \geq v$.

**Step 4.** Finally, we discuss how to find the optimal competitive ratio. The functions $\tau(\cdot)$, $b(\cdot)$, and $B(\cdot)$ defined above are all (implicitly) functions of the desired competitive ratio $\alpha$. To make this dependence explicit, let $B_\alpha(\cdot)$ denote the budget function for some fixed $\alpha$. It can be easily verified that $B_\alpha(v_{\max})$ is a continuous and monotonically increasing function of $\alpha$. Assume $U(v_{\max}) > 1$ since otherwise we can accept all items by setting $b(x) = u(x)$ and we obtain a competitive ratio of 1. We set $\alpha^* \in (0, 1)$ to be such that $B_{\alpha^*}(v_{\max}) = 1$. Again, by the intermediate value theorem, such an $\alpha^*$ is guaranteed to exist and further one can compute it up to arbitrary precision by binary search. For clarity, we include a formal description of the algorithm in the Supplementary Material.

### 3.1 Analysis

We begin by proving that our algorithm is indeed $\alpha^*$-competitive for any instance of C-KNAPSACK-FP that respects the given frequency predictions. Let $\tilde{v}$ be the largest value such that $A(\tilde{v}) = B(\tilde{v})$; if no such a value exists, set $\tilde{v} = v_{\min}$. Let ALG and OPT denote the total value of items chosen by our online algorithm and the optimal solution respectively. In the following two claims, we first bound ALG and OPT respectively.

**Claim 3.** $\mathrm{ALG} \geq \int_{x=v_{\min}}^{\tilde{v}} x \cdot b(x)\,\mathrm{d}x + \int_{x=\tilde{v}}^{v_{\max}} x \cdot s(x)\,\mathrm{d}x.$

*Proof.* For convenience of notation, let $a(v) = \frac{\mathrm{d}}{\mathrm{d}x}A(x)|_{x=v}$ denote the marginal amount of items of value $v$ accepted by the algorithm. Since we have $A(v) < B(v)$ for all $v > \tilde{v}$, the algorithm accepts all items having value larger than $\tilde{v}$ and thus $a(x) = s(x), \forall x > \tilde{v}$. Also, by definition of $\tilde{v}$, we have $A(\tilde{v}) = B(\tilde{v})$. Thus, we have the following.

$$\mathrm{ALG} = \int_{x=v_{\min}}^{v_{\max}} x \cdot a(x)\mathrm{d}x = \int_{x=v_{\min}}^{\tilde{v}} x \cdot a(x)\mathrm{d}x + \int_{x=\tilde{v}}^{v_{\max}} x \cdot s(x)\mathrm{d}x$$

integrating the first term by parts yields

$$\mathrm{ALG} = \tilde{v} \cdot A(\tilde{v}) - v_{\min} \cdot A(v_{\min}) - \int_{v_{\min}}^{\tilde{v}} A(x)\mathrm{d}x + \int_{x=\tilde{v}}^{v_{\max}} x \cdot s(x)\mathrm{d}x$$

since $A(\tilde{v}) = B(\tilde{v})$ and $A(x) \leq B(x)$, $\forall x$, by definition of our algorithm

$$\geq \tilde{v} \cdot B(\tilde{v}) - v_{\min} \cdot B(v_{\min}) - \int_{v_{\min}}^{\tilde{v}} B(x) \mathrm{d}x + \int_{x=\tilde{v}}^{v_{\max}} x \cdot s(x) \mathrm{d}x$$

$$= \int_{x=v_{\min}}^{\tilde{v}} x \cdot b(x) \mathrm{d}x + \int_{x=\tilde{v}}^{v_{\max}} x \cdot s(x) \mathrm{d}x. \qquad \square$$

**Claim 4.** OPT $\leq$ OPT$(\mathcal{M}(\tilde{v})) + \int_{x=\tilde{v}}^{v_{\max}} x \cdot (s(x) - \ell(x)) \, \mathrm{d}x$.

*Proof.* Let the optimum solution accept $s(x) - \ell(x)$ items of value $x$ for free without consuming the knapsack capacity for all $x \in [\tilde{v}, v_{\max}]$. The remaining items form an instance $\mathcal{I}'$ that is dominated by $\mathcal{M}(\tilde{v})$, i.e., for any value $x \in [v_{\min}, v_{\max}]$, the amount of items having value $x$ in the instance $\mathcal{I}'$ is at most the amount of items with value $x$ in $\mathcal{M}(\tilde{v})$. However, by definition, OPT$(\mathcal{M}(\tilde{v}))$ is the maximum profit that one can obtain from $\mathcal{M}(\tilde{v})$ and the claim follows. $\qquad \square$

The following is from the definition of SENTINEL and is useful to relate the value of the optimal solution for any instance of the form $\mathcal{M}(v)$ to the budget function $b(\cdot)$ utilized by our algorithm.

**Claim 5.** *For any value $v \in [v^*, v_{\max}]$, $\alpha^* \cdot$ OPT$(\mathcal{M}(v)) = \int_{x=v}^{v_{\max}} x \cdot \ell(x) \, \mathrm{d}x + \int_{x=v_{\min}}^{v} x \cdot b(x) \, \mathrm{d}x$.*

*Proof.* For all $v \geq v^*$, we have the following.

$$\alpha^* \cdot \text{OPT}(\mathcal{M}(v)) = \alpha^* \cdot \left( \text{OPT}(\mathcal{M}(v^*)) + \int_{x=v^*}^{v} \frac{\mathrm{d}}{\mathrm{d}x} \text{OPT}(\mathcal{M}(x)) \mathrm{d}x \right)$$

substituting (1) and (2),

$$= \int_{x=v^*}^{v_{\max}} x \cdot \ell(x) \mathrm{d}x + \int_{x=v^*}^{v} x \cdot \tau(x) \mathrm{d}x.$$

The claim follows by the definition of $b(x) = \ell(x) + \tau(x)$ for all $x \geq v^*$, and $b(x) = 0$, $\forall x < v^*$. $\quad \square$

Finally, we are ready to show that the algorithm is indeed $\alpha^*$-competitive. Claims 3 and 5 together yield that:

$$\text{ALG} \geq \alpha^* \text{OPT}(\mathcal{M}(\tilde{v})) + \int_{x=\tilde{v}}^{v_{\max}} x \cdot (s(x) - \ell(x)) \mathrm{d}x$$

$$\geq \alpha^* \cdot \left( \text{OPT}(\mathcal{M}(\tilde{v})) + \int_{x=\tilde{v}}^{v_{\max}} x \cdot (s(x) - \ell(x)) \mathrm{d}x \right) \geq \alpha^* \cdot \text{OPT},$$

where the last inequality follows from Claim 4. Thus, we have the following theorem. As discussed, the value of $\alpha^*$ depends on the predictions.

**Theorem 6.** *For any input instance that respects the frequency predictions $\{\ell(v), u(v)\}_{v \in [v_{\min}, v_{\max}]}$, the SENTINEL algorithm is $\alpha^*$-competitive for C-KNAPSACK-FP.*

### 3.1.1 Optimality of Competitive Ratio

In this section we show that the competitive ratio obtained by SENTINEL is optimal. In other words, no algorithm, even randomized, can obtain a competitive ratio of better than $\alpha^*$ for C-KNAPSACK-FP.

We will consider the family of instances $\{\mathcal{M}(v)\}_{v \in [v_{\min}, v_{\max}]}$. Recall that in the instance $\mathcal{M}(v)$, first $\ell(x)$ amount of items of all values $x \in [v_{\min}, v_{\max}]$ arrive in non-decreasing order of value, followed by $u(x) - \ell(x)$ amount of items of value $x \in [v_{\min}, v]$, which also arrive in non-decreasing order of value. Consequently, until an algorithm sees items of value higher than $v$ in the second stage, it cannot distinguish between instances in $\{\mathcal{M}(x)\}_{x \in [v, v_{\max}]}$. In other words, the instance $\mathcal{M}(v)$ is a *prefix* of each of the instances in $\{\mathcal{M}(x)\}_{x \in [v, v_{\max}]}$. Thus for any value $v$, the adversary can either strategically stop the instance at $\mathcal{M}(v)$ or continue to release higher-valued items.

**Lemma 7.** *For any given set $\{\ell(v), u(v)\}_{v \in [v_{\min}, v_{\max}]}$ of frequency predictions, let $\alpha^*$ denote the competitive ratio of the SENTINEL algorithm. Then no deterministic algorithm can obtain a competitive ratio larger than $\alpha^*$ on all instances in the family $\{\mathcal{M}(v)\}_{v \in [v_{\min}, v_{\max}]}$.*

To see why randomization does not help, consider the deterministic algorithm that accepts each item by the expected amount that the optimum randomized algorithm accepts. This is well-defined as we are allowed to choose items fractionally. It is easy to see that the algorithm never exceeds the knapsack capacity and obtains as much value as the randomized algorithm. Thus, we have:

**Theorem 8.** *For any given set $\{\ell(v), u(v)\}_{v \in [v_{\min}, v_{\max}]}$ of frequency predictions, there exists no randomized algorithm that attains a competitive ratio better than that of the* SENTINEL *algorithm.*

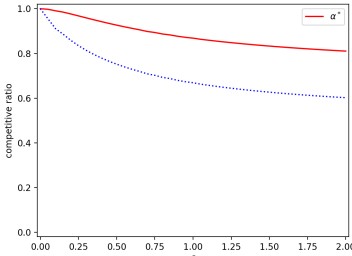

Figure 1: Illustration of the optimal competitive ratio $\alpha^*$ obtained by SENTINEL (in red) for predictions $\ell_v = 1/100$ and $u_v = (1 + \delta) \cdot \ell_v$ for integers $v \in [v_{\min} = 1, v_{\max} = 100]$. The competitive ratio is shown for $\delta \in [0, 2]$. For comparison, the competitive ratio resulting from accepting $\ell_v$ items of value $v$ is shown in blue, i.e., $\text{OPT}(\mathcal{M}(0))/\text{OPT}(\mathcal{M}(v_{\max}))$.

Thus SENTINEL yields the *optimal* competitive ratio for any given set of predictions. It can be easily verified that the classical adversarial online knapsack problem is equivalent to the special case with predictions $\ell(v) = 0$ and $u(v) = \infty$ for all $v \in [v_{\min}, v_{\max}]$. For this special case, we recover the competitive ratio $\frac{1}{1+\ln(\frac{v_{\max}}{v_{\min}})}$ of ZCL. For more informative predictions, we obtain much tighter competitive ratios. Indeed, if $\ell(v) = u(v), \forall v \in [v_{\min}, v_{\max}]$, then $\alpha^* = 1$ and SENTINEL obtains the optimum solution. Figure 1 illustrates the optimum competitive ratio for a family of predictions.

## 3.2 Need for Sentinel

Prior algorithms for (variants of) the online knapsack problem [31, 30] have all been *threshold* algorithms, i.e., the algorithm chooses a threshold $\Psi$ that is a function of the amount of residual capacity left in the knapsack and the algorithm accepts an item if and only if its value is higher than the threshold. On the other hand, the SENTINEL algorithm does not fit into this framework since the threshold for choosing an item depends on the values of items already chosen by the algorithm. In this section we show that there are predictions $\{l_v; u_v\}_{v \in V}$ such that no pure *threshold-based* algorithm can achieve the best competitive ratio. In fact, these predictions use only two values.

Let $\mathcal{I}$ denote a family of instances that respect the following predictions. There are two distinct values, $V = \{1, 2\}$. The predicted bounds are given by $l_1 = l_2 = 0$ and $u_1 = u_2 = 2/3$. As discussed in Section 3.1.1, since randomization has no benefits, we only consider deterministic algorithms.

**Claim 9.** *No deterministic online algorithm that maintains a non-decreasing threshold function $\Psi(z)$, where $z$ is the knapsack capacity used, and accepts an item if and only if its value is higher than $\Psi(z)$ has a competitive ratio better than $4/5$ for the family of instances $\mathcal{I}$.*

On the other hand, with these predictions there is a simple algorithm that obtains a competitive ratio of $6/7$. Let $\mathcal{A}_{\text{sentinel}}$ denote the following algorithm: accept all items of value 2 and accept at most $4/7$ items of value 1. The proof of the following claim follows from a simple case analysis.

**Claim 10.** *For the instances in $\mathcal{I}$, the algorithm $\mathcal{A}_{\text{sentinel}}$ yields a competitive ratio of $6/7$.*

## 4 Generalized One-Way Trading

In the classical one-way trading problem [13], a trader needs to convert a unit of money from one currency to another. Faced with a sequence of (unknown) exchange rates that are presented online, the goal is to maximize the amount of money obtained in the target currency. In this section we consider the following generalized one-way trading problem, also considered by [30]: There are $T$ different types of items and each type $t$ is associated with a *concave* value function $f_t$. For convenience, we consider the setting where each item $i$ has the same infinitesimal size $\lambda$. Items arrive online and the total value earned by the algorithm is $\sum_t f_t(x_t)$, where $x_t$ denotes the total size of items of type $t$

accepted by the algorithm. The goal is to design an algorithm that maximizes the total value such that the total size of accepted items is at most one. As in the online knapsack problem, we assume that the algorithm has access to a lower bound $\ell_t$ and an upper bound $u_t$ on the total size of items of type $t$.

We now show that we can reduce this problem to the online knapsack problem with predictions. For each type $t$, we create $\ell_t/\lambda$ *mandatory* items with values $f_t(\lambda), f_t(2\lambda) - f_t(\lambda), \ldots, f_t(\ell_t) - f_t(\ell_t - \lambda)$ respectively. Additionally, we create $u_t/\lambda$ *optional* items with values $f_t(\ell_t + \lambda) - f_t(\ell_t), \ldots, f_t(u_t) - f_t(u_t - \lambda)$ respectively. Note that many created items belonging to different types may have the same value. For each value $v$, let $\ell(v)$ be the total size of *mandatory* items of value $v$ and similarly let $u(v)$ be the total size of all *mandatory* and *optional* items of value $v$. Let $\mathcal{A}$ be an online algorithm for the knapsack problem using predictions $\{\ell(v), u(v)\}$. For the generalized one-way trading problem, as items arrive online, we construct an instance of the knapsack problem online such that the $k$th item of type $t$ has value $f_t(k\lambda) - f_t((k-1)\lambda)$. Finally, we accept an item (in the generalized one-way trading problem) iff the corresponding item is accepted by algorithm $\mathcal{A}$ in the knapsack instance.

It is easy to verify that the optimum solution attains the same value, say OPT, in both the problem instances. Let $\alpha^*$ denote the competitive ratio attained by $\mathcal{A}$, and hence the total value obtained by $\mathcal{A}$ on the knapsack instance is at least $\alpha^*$OPT. On the other hand, suppose $\mathcal{A}$ accepts $k$ items of type $t$, then due to concavity of $f_t$, it attains a value of at most $\sum_{j=1}^{k} f_t(j\lambda) - f_t((j-1)\lambda) = f_t(k\lambda)$. Thus, the total value obtained by the online algorithm for the generalized one-way trading problem is at least that obtained by $\mathcal{A}$ and hence it is also $\alpha^*$-competitive.

# 5 Two-Stage Knapsack

Consider the following two-stage version of the typical fractional online knapsack problem. An instance here is parameterized by a discount factor $0 \le \lambda \le 1$. Once again, for convenience, we work in the continuous model, so there is a continuum of items each having infinitesimal size[iv]. We assume that all items have a value between $[v_{\min}, v_{\max}]$. When an item of value $v$ arrives, an algorithm can either pick the item and receive a value of $v$ or it may choose to reject the item. After all items have arrived, the second stage starts and the algorithm can go back to previously rejected items and choose some of them, subject to the knapsack capacity. However, an item originally of value $v$ is only worth $\lambda v$ in the second stage. Note that an item cannot be rejected once it gets accepted in either stage.

We note that this two-stage knapsack formulation naturally interpolates between the online and offline versions of the standard knapsack problem; when $\lambda = 0$, an algorithm cannot obtain any value from the second stage and hence reduces to the regular online knapsack problem, whereas with $\lambda = 1$, an algorithm can commit to only accepting items in the second stage after all items have been revealed and thus the problem reduces to the offline version.

To the best of our knowledge this problem has not been considered in the literature. Thus we first show how to obtain the optimal competitive ratio for the problem in the absence of predictions. We then continue to show how to incorporate predictions on item frequencies for each value. We present the algorithm without predictions and defer the one with predictions to the Supplementary Material.

## 5.1 Algorithm for Two-Stage Knapsack without Frequency Predictions

As before let $\alpha^*$ be the optimal competitive ratio for the problem, which will be decided later. In the absence of predictions, the optimal competitive ratio can be attained by a *threshold* algorithm. In the first stage of the problem, the algorithm will maintain a threshold function depending on the knapsack capacity used and accept an item if and only if its value is higher than the current threshold. Finally, in the second stage, it is easy to observe that any reasonable algorithm accepts the highest value items that can still be accommodated in the knapsack. We now show how to systematically derive such an algorithm.

For any value $v$, let $z(v)$ be the fraction of the knapsack capacity for which the algorithm maintains the threshold at value $v$. To derive $z(v)$ we will consider the following family of adversarial instances: (i) items arrive in increasing order of value; (ii) there are enough items of each value to fill the entire knapsack; and (iii) the adversary can stop releasing items at any point in time.

---

[iv]As in Lemma 2, when items have sufficiently small sizes, this assumption only leads to a small additive loss in the competitive ratio.

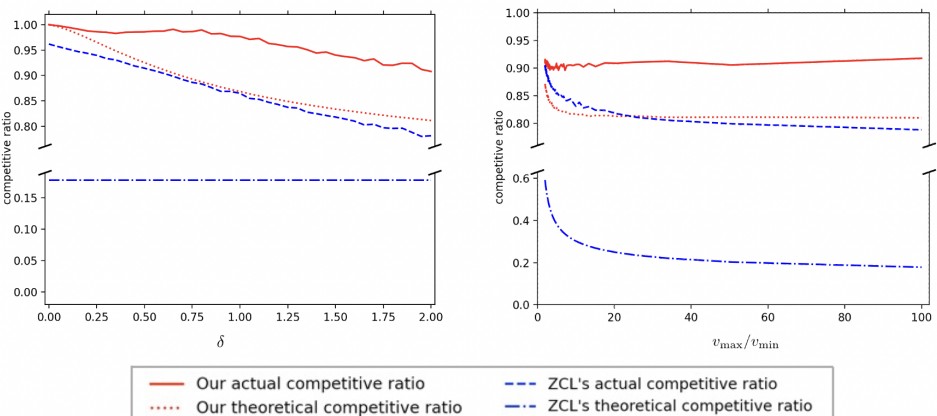

Figure 2: Illustration of the empirical and theoretical competitive ratios of our algorithm SENTINEL and ZCL. In the left, $v_{\min} = 1$, $v_{\max} = 100$, and $\delta \in [0, 2]$; and in the right, $v_{min} \in [1, 100]$, $v_{\max} - v_{\min} = 99$ and $\delta = 2$. In both $u_v = \lceil (1 + \delta) \rceil \ell_v$. All items have size 1/10000.

We construct $z(v)$ that is continuous at all $v \in (v_{\min}, v_{\max}]$ as follows. First we set, $z(v_{\min}) = \frac{\alpha^* - \lambda}{1 - \lambda}$. This is because if the adversary only gives items of value $v_{\min}$ and they are enough to fill up the knapsack, then by accepting $z(v_{\min})$ of them in the first stage and more to the full capacity in the second stage we obtain profit $(z(v_{\min}) + \lambda(1 - z(v_{\min})))v_{\min} = \alpha^* v_{\min}$.

Now consider any value $v > v_{\min}$. If the adversary stops the instance at value $v$, the profit obtained by the algorithm is:

$$\text{ALG}(v) := v_{\min} z(v_{\min}) + \int_{x \in (v_{\min}, v]} x \cdot z(x) \mathrm{d}x + \lambda v \Big(1 - z(v_{\min}) - \int_{x \in (v_{\min}, v]} z(x) \mathrm{d}x\Big).$$

Here, the first two terms denote the value earned by the algorithm in the first stage while the last term denotes the profit gained in the second stage. In particular, $(1 - z(v_{\min}) - \int_{x=v_{\min}}^{v} z(x) \mathrm{d}x)$ is the amount of capacity left in the knapsack after the first stage and the algorithm will fill it with the best items—of value $v$—and receive value of $\lambda v$ per each unit.

To maintain a competitive ratio of $\alpha^*$, we need to maintain for all $v \in (v_{\min}, v_{\max}]$, $\mathcal{M}(v) = \alpha^* v$. By solving this, together with $z(v_{\min}) + \int_{x \in (v_{\min}, v_{\max}]} z(x) \mathrm{d}x = 1$, we can find $\alpha^*$ and $z(v)$. Then, set the threshold function $\Psi(t)$ as follows:

$$\Psi(t) = \begin{cases} v_{\min} & \text{if } 0 \leq t \leq \frac{\alpha^* - \lambda}{1 - \lambda} \\ Z^{-1}(t) & \text{otherwise} \end{cases},$$

where $Z(v) := \int_{x=v_{\min}}^{v} z(x) \mathrm{d}x$; this integral includes $z(v_{\min})$. And if we see an item of value $v$ in the first phase when the knapsack is $t$-full, we accept it if $v \geq \Psi(t)$. In other words, we keep the threshold at $v_{\min}$ until the knapsack gets $\frac{\alpha^* - \lambda}{1 - \lambda}$-full; afterwards, keep the threshold at $v$ for $z(v)$ units. In the second phase, we accept the highest valued remaining items to the full capacity. Further details are in the Supplementary Material.

## 6 Experiments

We design experiments to show that our algorithm, SENTINEL, achieves a considerably larger profit than ZCL [31] even when we are given loose frequency bounds as predictions. Specifically, we consider the following setup. The knapsack capacity is set to 1. Each item is assumed to have a size $10^{-4}$ and an integer value in the range of $[v_{\min} = 1, v_{\max} = 100]$. For each value $v$, its frequency lower bound $\ell_v$ is sampled from $[50, 150]$ independently and uniformly at random; therefore, in expectation, at least $10^4$ items arrive in an instance. We use a control parameter $\delta$ to derive the upper bounds $u_v$ from $\ell_v$ and set $u_v$ to $\lceil (1 + \delta)\ell_v \rceil$. For each instance, $s_v$ is sampled from $[\ell_v, u_v]$

independently and uniformly at random and items arrive in random order. The results are the geometric average over 10 instances.

In the first experiment we observe how the competitive ratios of SENTINEL and ZCL change as we vary $\delta$ from 0 to 2. Intuitively, smaller $\delta$ value means better predictions. We confirm that SENTINEL outperforms ZCL for all values of $\delta$ in terms of empirical as well as theoretical competitive ratios. The actual competitive ratios degrade as $\delta$ grows, but SENTINEL continues to dominate ZCL.

In the second experiment, we consider different values of $v_{\min}$ in the range of $\{1, \ldots, 100\}$, fixing the difference $v_{\max} - v_{\min} = 99$ and $\delta = 2$. The goal of this experiment is to see how the competitive ratios change as the ratio $v_{\max}/v_{\min}$ changes. Note that the ratio decreases from 100 to nearly 2. Recall that ZCL has competitive ratio $1/(1 + \ln(v_{\max}/v_{\min}))$. Again, SENTINEL consistently outperforms ZCL, although the difference becomes smaller as $v_{\max}/v_{\min}$ decreases.

## 7 Conclusions

In this paper we studied the online knapsack problem and its extensions when given the range of the number of items of each value as predictions. We showed that even such weak predictions can be exploited to give provably better competitive ratios. It would be interesting to revisit other problems with such weak predictions. Another fruitful direction is to explore other types of weak predictions.

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
