## Supplementary Material

## A  Missing Proofs

**Lemma 2.** *Given a prediction $P = \{\ell_v; u_v\}_{v \in V}$ for* KNAPSACK-FP, *we can create a prediction $P' = \{\ell(v); u(v)\}_{v \in [v_{\min}, v_{\max}]}$ for* C-KNAPSACK-FP *in polynomial time such that*

1. *An $\alpha$-competitive algorithm for* C-KNAPSACK-FP *with prediction $P'$ implies an $(\alpha - \frac{v_{\max}}{v_{\min}} \cdot |V| \cdot \max_i s_i)$-competitive algorithm for* KNAPSACK-FP *with prediction $P$.*
2. *If no algorithm has a competitive ratio better than $\alpha$ for* C-KNAPSACK-FP *with prediction $P'$, then no algorithm has a competitive ratio better than $\alpha + \max_i s_i$ for* KNAPSACK-FP *with prediction $P$.*

*Proof.* We first describe how to create $P'$ from $P$. We define a continuous piecewise- linear function $\ell(x)$ as follows. Consider each $v \in V$. Let $\delta$ be infinitesimally small. For $x \in [v - \delta, v + \delta]$, let $\ell(v - \delta) = \ell(v + \delta) = 0$, $\ell(v) = \frac{\ell_v}{\delta}$ and interpolate the function values at $v - \delta, v, v + \delta$ by linear functions. It is an easy exercise to show that $\int_{x=v-\delta}^{v+\delta} \ell(x) \mathrm{d}x = \ell_v$ and $\lim_{\delta \to 0} \int_{x=v-\delta}^{v+\delta} x \ell(x) \mathrm{d}x = v \ell_v$. We analogously define $u(x)$ such that $u(v - \delta) = u(v + \delta) = 0$, $u(v) = \frac{u_v}{\delta}$. By construction, we have that $u(x)/\ell(x)$ has the same value for all $x \in (v - \delta, v + \delta)$; thus $\frac{\ell(x)}{\ell_v} = \frac{u(x)}{u_v}$ for all $x \in [v - \delta, v + \delta]$. For all the other $x$ values, let $\ell(x) = 0$. Note that $\ell(x) = 0$ for all $x \notin [v_{\min} - \delta, v_{\max} + \delta]$. Define $u(x)$ similarly. It is easy to see that $\ell(x) \leq u(x)$ by definition.

To show the first claim, consider any $\alpha$-competitive algorithm $A'$ for C-KNAPSACK-FP with prediction $P'$. Consider any instance $I$ respecting $P$. An item $i$ of size $s_i$ and profit $p_i$ in $I$ is converted into $s_i$ items of value approximately equal to $v = p_i/s_i$ in $I'$. More precisely, we create $\frac{s_i}{\ell_v} \ell(x)$ items of value $x$ for each $x \in [v - \delta, v + \delta]$. Then, the resulting instance $I'$ respects $P'$. This is because $\frac{s_v}{\ell_v} \ell(x)$ items of value $x$ are created and $\frac{s_v}{\ell_v} \ell(x) \geq \ell(x)$, meaning that the lower bound frequency predictions are respected. Similarly, as observed before, $\frac{s_v}{\ell_v} \ell(x) = \frac{s_v}{u_v} u(x) \leq u(x)$.

Suppose $A'$ has accepted $a'(v)$ items of value approximately equal to $v \in V$—more precisely, in $[v - \delta, v + \delta]$ for some $v \in V$. We define an algorithm $A$ for KNAPSACK-FP trying to keep up with $A'$. When an item $i$ of value $v$ arrives in $I$, corresponding items are created in $I'$ and algorithm $A'$ updates $a'$. Then, $A$ accepts item $i$ if it will have accepted at most $a'(v)$ items of value $v$ after accepting it.

How much can $A$ be behind $A'$? Let $a(v)$ be the total size of items of value $v$ accepted by $A$. It is an easy exercise to see that $a'(v) - s_{\max} \leq a(v) \leq a'(v)$, where $s_{\max} := \max_j s_j$ is the maximum item size in $I$. When $\delta \to 0$, $A'$ gets $a'(v)v$ profit while as $A$ gets $a(v)v$ profit. Thus, $A$ can be behind $A'$ by at most $\sum_{v \in V} v s_{\max} \leq v_{\max} \cdot |V| \cdot s_{\max}$ in terms of profit. Clearly, the optimum is at least $v_{\min}$, which implies an additive loss of at most $\frac{v_{\max}}{v_{\min}} \cdot |V| \cdot s_{\max}$ in the competitive ratio.

We shift to showing the second claim. We use the same $P'$ and $I'$ as above. For the sake of contradiction, suppose there exists an algorithm $A$ for KNAPSACK-FP that has a competitive ratio of $\beta > \alpha + s_{\max}$. Consider the following algorithm $A'$ for C-KNAPSACK-FP: If $A$ accepts an item of value $v$ and size $s$, we let $A'$ accept the corresponding items. It is easy to see that $A$ and $A'$ achieve the same profit $Q$ at the end as $\delta \to 0$. But the optimum solution for $I'$ has more profit than the optimum solution for $I$ because it can accept items fractionally. The optimum solution for $I'$ accepts items from the highest value until it saturates the knapsack capacity. Thus, the corresponding solution for $I$ accepts items integrally, possibly except one item of some value $v'$. Therefore, $\text{OPT}' - v' s_{\max} \leq \text{OPT} \leq \text{OPT}'$ and $v' \leq \text{OPT}'$ (when $\delta \to 0$), where OPT and OPT$'$ denote the optimum for $I$ and $I'$, respectively. By definition, $\beta = \frac{Q}{\text{OPT}}$. Since $\frac{Q}{\text{OPT}} - \frac{Q}{\text{OPT}'} \leq \frac{Q \cdot v' s_{\max}}{\text{OPT} \cdot \text{OPT}'} \leq s_{\max}$, $A$'s competitive ratio is at least $\beta - s_{\max}$, a contradiction. $\square$

**Lemma 7.** *For any given set $\{\ell(v), u(v)\}_{v \in [v_{\min}, v_{\max}]}$ of frequency predictions, let $\alpha^*$ denote the competitive ratio of the* SENTINEL *algorithm. Then no deterministic algorithm can obtain a competitive ratio larger than $\alpha^*$ on all instances in the family $\{\mathcal{M}(v)\}_{v \in [v_{\min}, v_{\max}]}$.*

*Proof.* Let $\mathcal{A}$ denote an arbitrary fixed deterministic algorithm. Abusing notation slightly, let $A(v)$ denote the total amount of items of value at most $v$ that are accepted by the algorithm $\mathcal{A}$ and let $a(v) = \frac{\mathrm{d}}{\mathrm{d}x}A(x)|_{x=v}$. Let $v_1$ be the smallest value $v$ such that $B(v) \neq A(v)$.[v] If there is no such value, then consider the adversarial instance $\mathcal{M}(v_{\max})$. By Claim 5, it follows that the algorithm $\mathcal{A}$ has competitive ratio exactly $\alpha^*$. Thus assuming that the value $v_1$ exists, we now consider two cases.

**Case 1.** $A(v_1) < B(v_1)$. In this case, the adversary stops the instance at $\mathcal{M}(v_1)$. Note that $v_1 > v^*$ since $B(v^*) = 0$ and $B(v)$ is non-decreasing in $v$. Since the algorithm accepted $a(x) = b(x)$ amount of items for all values less than $v_1$, less than $b(v_1)$ items of value $v_1$, and at most $l(v)$ items of each value $v > v_1$, the algorithm's profit is upper bounded by

$$\int_{x=v_{\min}}^{v_1} x \cdot a(x)\mathrm{d}x + \int_{x=v_1}^{v_{\max}} x \cdot \ell(x)\mathrm{d}x < \int_{x=v_{\min}}^{v_1} x \cdot b(x)\mathrm{d}x + \int_{x=v_1}^{v_{\max}} x \cdot \ell(x)\mathrm{d}x = \alpha^* \cdot \mathrm{OPT}(\mathcal{M}(v_1)),$$

where the last equality follows from Claim 5. Thus, the algorithm fails to achieve the competitive ratio $\alpha^*$.

**Case 2.** $A(v_1) > B(v_1)$. Recall that by definition of $\alpha^*$, we have $B(v_{\max}) = 1$.[vi] As the knapsack capacity is 1, we must have $A(v_{\max}) \leq 1$. Thus, there must exist $v > v_1$ such that $A(v) = B(v)$. Let $v_2$ be the minimum among all such values $v$. Note that $v_2 > v^*$ since $B(v^*) = 0$ and $B(v_2) = A(v_2) \geq A(v_1) > B(v_1) \geq 0$. In this case, the adversary stops at $v_2$ declaring $\mathcal{M}(v_2)$ as the final instance. By definition of $v_2$, we have $A(v_2) = B(v_2)$ and $A(x) \geq B(x)$ for all $x \leq v_2$. Using integration by parts we have:

$$\int_{x=v_{\min}}^{v_2} x \cdot a(x)\mathrm{d}x = v_2 \cdot A(v_2) - v_{\min} \cdot A(v_{\min}) - \int_{x=v_{\min}}^{v_2} A(x)\mathrm{d}x$$

$$\leq v_2 \cdot B(v_2) - v_{\min} \cdot B(v_{\min}) - \int_{x=v_{\min}}^{v_2} B(x)\mathrm{d}x$$

$$= \int_{x=v_{\min}}^{v_2} x \cdot b(x)\mathrm{d}x.$$

As before, the total profit of algorithm $\mathcal{A}$ is at most:

$$\int_{x=v_{\min}}^{v_2} x \cdot a(x)\mathrm{d}x + \int_{x=v_2}^{v_{\max}} x \cdot \ell(x)\mathrm{d}x \leq \int_{x=v_{\min}}^{v_2} x \cdot b(x)\mathrm{d}x + \int_{x=v_1}^{v_{\max}} x \cdot \ell(x)\mathrm{d}x = \alpha^* \cdot \mathrm{OPT}(\mathcal{M}(v_2)).$$

Thus, in all cases, we have shown that no deterministic algorithm can have a competitive ratio better than $\alpha$. $\qquad\square$

**Claim 9.** *No deterministic online algorithm that maintains a non-decreasing threshold function $\Psi(z)$, where $z$ is the knapsack capacity used, and accepts an item if and only if its value is higher than $\Psi(z)$ has a competitive ratio better than $4/5$ for the family of instances $\mathcal{I}$.*

*Proof.* Since there are only two distinct item values, assume w.l.o.g. that we have the following threshold function for some parameter $\lambda$:

$$\Psi(t) = \begin{cases} 1 & \text{if } t \in [0, \lambda] \\ 2 & \text{if } t \in [\lambda, 1]. \end{cases}$$

We begin by considering the most interesting case when $\lambda \in [1/3, 2/3]$. Suppose $\lambda$ items of value 2 first arrive, and then, $2/3$ items of value 1 arrive. Then, the algorithm gets profit $2\lambda$ as it accepts no item of value 1. On the other hand, the optimum solution can obtain profit $2\lambda + 1 \cdot (1 - \lambda) = 1 + \lambda$. The competitive ratio is upper bounded by $\frac{2\lambda}{1+\lambda} \leq 4/5$ where the equality holds when $\lambda = 2/3$.

---

[v] Strictly speaking, this may not exist in general in the continuous setting, but we assume wlog that values are sufficiently discretized to avoid such issues.

[vi] The only case that this it not true is that the knapsack is big enough to accept all the items, in which case our algorithm has competitive ratio 1.

Next, suppose $\lambda \in [0, 1/3]$. If only items of value 1 arrive, by 2/3 amount, the algorithm gets profit 1/3, whereas the optimum profit is 2/3. Thus, in this case, the competitive ratio is only 1/2.

Finally, consider the case when $\lambda \in [2/3, 1]$. In this case, 2/3 amount of items of each value arrive in non-decreasing order of their value. It is an easy exercise to see that the algorithm obtains profit $1 \cdot (2/3) + 2 \cdot (1/3) = 4/3$, but the optimum is 5/3. Again, we only obtain a competitive ratio of 4/5. $\qquad\square$

**Claim 10.** *For the instances in $\mathcal{I}$, the algorithm $\mathcal{A}_{sentinel}$ yields a competitive ratio of $6/7$.*

*Proof.* We consider the following algorithm: if the item's value is 2, always accept it; otherwise, we accept up to 4/7 items of value 1. Let $A$ and OPT denote the profit achieved by the algorithm and the optimum, respectively. Suppose $x$ items of value 1 and $y$ items of value 2 arrive. Then, we have,

$$\text{OPT} = 2 \cdot y + 1 \cdot \min\{1 - y, x\}.$$

Clearly, the algorithm can suffer the most when given items of value 1 first. Then, we have

$$A = 2 \cdot \min\{y, 1 - \min\{x, 4/7\}\} + 1 \cdot \min\{x, 4/7\}.$$

Note that $x, y \in [0, 2/3]$ in the instances that respect the given predictions.

We consider the following cases:

1. $x + y \geq 1$: In this case note that $x, y \in [1/3, 2/3]$.
   (a) $x \geq 4/7$:
      i. $y \geq 3/7$:
      
      $$\frac{A}{\text{OPT}} = \frac{1 \cdot 4/7 + 2 \cdot 3/7}{1 + y} \geq \frac{10/7}{5/3} = 6/7, \text{ where the equality holds when } y = 2/3.$$

      ii. $y < 3/7$:
      $$\frac{A}{\text{OPT}} = \frac{1 \cdot 4/7 + 2 \cdot y}{1 + y} \geq \frac{4/7 + 2/3}{4/3} = 13/14,$$

      where the last inequality follows from $y \geq 1/3$, which is the case because $x \in [0, 2/3]$ due to the given prediction.
   (b) $x < 4/7$:
      i. $y \geq 1 - x$:
      $$\frac{A}{\text{OPT}} = \frac{x + 2(1 - x)}{1 + y} \geq \frac{2 - x}{1 + 2/3} \geq \frac{2 - 4/7}{5/3} = 6/7.$$

      ii. $y < 1 - x$: The inequalities are contradictory.
2. $x + y < 1$:
   (a) $x \geq 4/7$:
      i. $y \geq 3/7$: The inequalities are contradictory.
      ii. $y < 3/7$:
      $$\frac{A}{\text{OPT}} = \frac{1 \cdot 4/7 + 2y}{x + 2y} \geq \frac{4/7}{x} \geq \frac{4/7}{2/3} \geq 6/7$$

   (b) $x < 4/7$:
      i. $y \geq 1 - x$: The inequalities are contradictory.
      ii. $y < 1 - x$:
      $$\frac{A}{\text{OPT}} = \frac{x + 2y}{x + 2y} = 1.$$

In all cases, we have $A/\text{OPT} \geq 6/7$, which establishes that the algorithm $A$ is $(6/7)$-competitive for the instance family $\mathcal{I}$. $\qquad\square$

## B SENTINEL Algorithm Description

We give a short summary of the SENTINEL algorithm here for clarity.

---

**Algorithm 1** Algorithm SENTINEL for C-KNAPSACK-FP

---

**Input:** $P = \{\ell(v); u(v)\}_{v \in [v_{\min}, v_{\max}]}$ as predictions

Compute $v_\alpha^*$, $\tau_\alpha(\cdot)$, $b_\alpha(\cdot)$, and $B_\alpha(\cdot)$, $\forall \alpha \in [0, 1]$ using Equations (1), (2), and (3)
Let $\alpha^*$ be such that $B_{\alpha^*}(v_{\max}) = 1$
Initialize $A(v) = 0$, $\forall v \in [v_{\min}, v_{\max}]$
**for** each item that arrives online **do**
    Let $v$ denote value of the item
    Let $s$ denote size of the item (infinitesimal)
    **if** $A(x) < B_{\alpha^*}(x), \forall x \geq v$ **then**
        Accept item
        Update $A(x) \leftarrow A(x) + s, \forall x \geq v$

---

## C   Algorithm for Items with Discrete Values

In this section we describe the discrete version of our algorithm SENTINEL for KNAPSACK-FP, while discussing some implementation issues. Recall that an item is assumed to have a value in $V = \{v_{\min} = v_0, v_1, \ldots, v_{k-1} = v_{\max}\}$. For simplicity, we will assume for a while that items can be accepted fractionally; this assumption will be removed at the end of this section. If an item is accepted fractionally, a partial profit is obtained in proportion to the fraction of the item accepted. As before, we are given $l_v \leq u_v$ for each $v \in V$ as prediction such that $s_v \in [l_v, u_v]$. For notational convenience, we will let $\ell_i, u_i, s_i$ denote $\ell_{v_i}, u_{v_i}, s_{v_i}$, respectively.

As the following steps directly correspond to those of SENTINEL in Section 3, we do not repeat the intuition behind them. Let $\mathcal{M}(v_i, \beta)$ be an instance such that $s_{i'} = \ell_{i'}$ for all $i' > i$, $s_i = \ell_i + \beta(u_i - \ell_i)$, and $s_{i'} = u_{i'}$ for all $i' < i$, where $\ell_{i'}$ items of each value $v_{i'}$ first arrive, and then $s_{i'} - \ell_{i'}$ items of each value $v_{i'}$ arrive in non-decreasing order of their value.

**Step 1.** Let $v_{i^*}$ denote the largest value in $V$ such that for some $\beta \in [0, 1]$, $(1 - \beta)\ell_{v_{i^*}} v_{i^*} + \sum_{i > i^*}^{k-1} \ell_{v_i} v_i = \alpha \cdot \mathrm{OPT}(\mathcal{M}(v_{i^*}, \beta))$. Let $\beta_{i^*}$ denote the value of $\beta$.

**Step 2.** Set $\tau_{i^*} v_{i^*} = \alpha(\mathrm{OPT}(\mathcal{M}(v_{i^*}, 1)) - \mathrm{OPT}(\mathcal{M}(v_{i^*}, \beta_{i^*}))$ for $i = i^*$ and $\tau_i v_i = \alpha(\mathrm{OPT}(\mathcal{M}(v_i, 1)) - \mathrm{OPT}(\mathcal{M}(v_{i-1}, 1)))$ for all $i > i^*$.

**Step 3.** For any value $i \in 0, 1, \ldots, k - 1$, we define a budget function as follows: $b_i = 0 \ \forall i < i^*$, $b_{i^*} = (1 - \beta_{i^*})\ell_{v_{i^*}} + \tau_{i^*}$, and $b_i = \ell_i + \tau_i \ \forall i > i^*$.

**Step 4.** Find the value of $\alpha^* \in (0, 1]$ by binary search such that $\sum_{k'=0}^{k-1} b_{k'} = 1$ to an arbitrary precision.

We accept an item of value $v$ if $A_i \leq B_i$ for all $i \in \{0, 1, \ldots, k-1\}$ after accepting it, where $A_x$ is the total size of items accepted so far. If items cannot be accepted fractionally, we may lose at most $s_{\max} v_i$ profit for each value $v_i \in V$ where $s_{\max}$ is the maximum item size. It is easy to see that the resulting competitive ratio is at least $\alpha^* - |V| \cdot s_{\max} \cdot v_{\max}/v_{\min}$ following the proof of Lemma 2.

Finally, we discuss how to maintain the invariant $A_i \leq B_i$ efficiently. Consider the following equivalent way of enforcing the invariant: Initially, set $b_j' = b_j$ for all $j \in \{0, 1, 2, \ldots, k-1\}$. When an item of value $v_i$ and size $s$ arrives, let $j = \arg \max_{i' \in [0, i]}(s \leq b_{i'}')$. If $j$ does not exist, then we reject the item. Otherwise, we accept it and update $b_j'$ to be $b_j' - s$. It is left as an exercise to verify the equivalence if items have infinitesimal sizes. Using a (balanced) binary search tree, we can handle each arriving item with $O(\log |V|)$ update time.

# D Two-stage Knapsack without Predictions

## D.1 Algorithm

As before let $\alpha^*$ be the optimal competitive ratio for the problem, which will be decided later. In the absence of predictions, the optimal competitive ratio can be attained by a *threshold* algorithm. In the first stage of the problem, the algorithm will maintain a threshold function depending on the knapsack capacity used and accept an item if and only if its value is higher than the current threshold. Finally, in the second stage, it is easy to observe that any reasonable algorithm accepts the highest value items that can still be accommodated in the knapsack. We now show how to systematically derive such an algorithm.

For any value $v$, let $z(v)$ be the fraction of the knapsack capacity for which the algorithm maintains the threshold at value $v$. To derive $z(v)$ we will consider the following family of adversarial instances: (i) items arrive in increasing order of value; (ii) there are enough items of each value to fill the entire knapsack; and (iii) the adversary can stop releasing items at any point in time.

We construct $z(v)$ that is continuous at all $v \in (v_{\min}, v_{\max}]$ as follows. First we set:

$$z(v_{\min}) = \frac{\alpha^* - \lambda}{1 - \lambda}. \tag{4}$$

This is because if the adversary only gives items of value $v_{\min}$ and they are enough to fill up the knapsack, then by accepting $z(v_{\min})$ of them in the first stage and more to the full capacity in the second stage we obtain profit $(z(v_{\min}) + \lambda(1 - z(v_{\min})))v_{\min} = \alpha^* v_{\min}$.

Now consider any value $v > v_{\min}$. If the adversary stops the instance at value $v$, the profit obtained by the algorithm is:

$$\text{ALG}(v) := v_{\min} z(v_{\min}) + \int_{x \in (v_{\min}, v]} x \cdot z(x) \mathrm{d}x + \lambda v \left(1 - z(v_{\min}) - \int_{x \in (v_{\min}, v]} z(x) \mathrm{d}x \right).$$

Here, the first two terms denote the value earned by the algorithm in the first stage while the last term denotes the profit gained in the second stage. In particular, $(1 - z(v_{\min}) - \int_{x=v_{\min}}^{v} z(x) \mathrm{d}x)$ is the amount of capacity left in the knapsack after the first stage and the algorithm will fill it with the best items—of value $v$—and receive value of $\lambda v$ per each unit.

To maintain a competitive ratio of $\alpha^*$, we need to maintain for all $v \in (v_{\min}, v_{\max}]$,

$$v_{\min} z(v_{\min}) + \int_{x \in (v_{\min}, v]} xz(x) \mathrm{d}x + \lambda v \left(1 - z(v_{\min}) - \int_{x \in (v_{\min}, v]} z(x) \mathrm{d}x \right) = \alpha^* v \tag{5}$$

differentiating w.r.t. $v$,

$$vz(v) + \lambda(1 - z(v_{\min})) - \lambda \left(vz(v) + \int_{x \in (v_{\min}, v]} z(x) \mathrm{d}x \right) = \alpha^*$$

rearranging,

$$vz(v) - \left(\frac{\lambda}{1 - \lambda}\right) \cdot \int_{x \in (v_{\min}, v]} z(x) \mathrm{d}x = \frac{\alpha^* - \lambda}{1 - \lambda} + \frac{\lambda z(v_{\min})}{1 - \lambda} \tag{6}$$

solving this ODE, for some $c$, we get

$$z(v) = cv^{\frac{2\lambda - 1}{1 - \lambda}} n.$$

To determine $\alpha^*$ and $c$, by taking $\lim_{v \to v_{\min}+}$ on (6), we have

$$c \, v_{\min}^{\frac{\lambda}{1 - \lambda}} = \frac{\alpha^* - \lambda}{1 - \lambda} + \frac{\lambda z(v_{\min})}{1 - \lambda} = \frac{\alpha^* - \lambda}{1 - \lambda} + \frac{\alpha^* - \lambda}{(1 - \lambda)^2}, \tag{7}$$

where the last equality follows from (4)

Further, we want to use the whole capacity. Thus, from $z(v_{\min}) + \int_{x \in (v_{\min}, v_{\max}]} z(x) \mathrm{d}x = 1$ and (4), we have

$$\frac{\alpha^* - \lambda}{1 - \lambda} + c \frac{1 - \lambda}{\lambda}(v_{\max}^{\frac{\lambda}{1 - \lambda}} - v_{\min}^{\frac{\lambda}{1 - \lambda}}) = 1 \tag{8}$$

We can determine the value of $\alpha^*$ and $c$ from (7) and (8). Here, we do not explicitly show their closed form as they are not very simple.

We are now ready to describe the algorithm. Set the threshold function $\Psi(t)$ as follows:

$$\Psi(t) = \begin{cases} v_{\min} & \text{if } 0 \leq t \leq \frac{\alpha^* - \lambda}{1-\lambda} \\ Z^{-1}(t) & \text{otherwise,} \end{cases},$$

where $Z(v) := \int_{x=v_{\min}}^{v} z(x)\mathrm{d}x$; this integral includes $z(v_{\min})$. And if we see an item of value $v$ in the first phase when the knapsack is $t$-full, we accept it if $v \geq \Psi(t)$. In other words, we keep the threshold at $v_{\min}$ until the knapsack gets $\frac{\alpha^* - \lambda}{1-\lambda}$-full; afterwards, keep the threshold at $v$ for $z(v)$ units. In the second phase, we accept the highest valued remaining items to the full capacity.

## D.2 Analysis

Suppose the last threshold value used by the algorithm was $y$. Let $A_1$ and $A_2$ denote the items we accepted in the first and second phases, respectively. Let $O$ be the items accepted by the optimum solution. As the algorithm accepts the highest-valued remaining items in the second phase, it must be that $A_2 \subseteq O \setminus A_1$. Let $v(P)$ denote the total profit of items in $P$ and $s(P)$ the total size of items in the same set. Then, the ratio of our algorithm's objective to the optimum is

$$\rho := \frac{v(A_1 \cap O) + v(A_1 \setminus O) + \lambda \cdot v(A_2)}{v(A_1 \cap O) + v(O \setminus A_1 \setminus A_2) + v(A_2)}.$$

For the sake of contradiction suppose $\rho < \alpha^*$. Observe that all items of value greater than $y$ are accepted by the algorithm in the first phase, and also by the optimum solution. To draw a contradiction we change the item values ensuring $\rho < \alpha^*$. First consider the items in $A_2$. By definition of $A_2$, all items in $A_2$ have value at most $y$. Due to the multiplier $\lambda$ of $v(A_2)$ in the numerator and the fact that $\lambda \leq \alpha^*$, we have

$$\frac{v(A_1 \cap O) + v(A_1 \setminus O) + \lambda y \cdot s(A_2)}{v(A_1 \cap O) + v(O \setminus A_1 \setminus A_2) + y \cdot s(A_2)} < \alpha^*,$$

which follows from the thought process of increasing the value of items in $A_2$ to $y$.

As no items in $O \setminus A_1$ have value greater than $y$, by increasing the value of the items in $O \setminus A_1$ to $y$, we can only decrease the ratio. Thus, we have,

$$\frac{v(A_1 \cap O) + v(A_1 \setminus O) + \lambda y \cdot s(A_2)}{v(A_1 \cap O) + y \cdot s(O \setminus A_1 \setminus A_2) + y \cdot s(A_2)} < \alpha^*,$$

Finally, we decrease the value of each item in $A_1$ to the threshold of our algorithm when it was accepted. As the items in $A_1 \cap O$ appear in both the numerator and denominator and the ratio is less than 1, decreasing the value of an item in $A_1 \cap O$ can only decrease the ratio. Further, it is clear that doing so for items in $A_1 \setminus O$ decreases the ratio as they only appear in the numerator. Finally, knowing that the threshold was always no greater than $y$, by increasing the value of the items in $A_1 \cap O$ only in the denominator, we have,

$$\frac{v_{\min} z(v_{\min}) + \int_{x \in (v_{\min}, v]} y \cdot z(y)\mathrm{d}x + \lambda y \cdot s(A_2)}{y \cdot s(A_1 \cap O) + y \cdot s(O \setminus A_1 \setminus A_2) + y \cdot s(A_2)} < \alpha^*.$$

The numerator and denominator are $\mathrm{ALG}(y)$ and $y$ respectively. Therefore, the ratio is exactly $\alpha^*$ due to (5), which is a contradiction.

## D.3 Optimality

In this section we show that no algorithm can have a competitive ratio better than $\alpha^*$. We closely follow the optimality proof in Section 3.1.1. We use the same adversarial instances as we used to compute $z(v)$ in Section 5.1. For easy reference, we restate them here: (i) all items arrive in increasing order of their value; (ii) there are enough items of each value to fill up the entire knapsack; and (iii) the adversary can stop releasing items at any point in time.

Fix any deterministic algorithm. Suppose it accepts $a(v)$ items of value $v$. Let $A(v)$ denote the total size of items of value at most $v$ accepted by the algorithm. If $A(v) = Z(v)$ for all $v$, then it coincides with our algorithm whose competitive ratio is $\alpha^*$. So, let $v_1 := \arg\min_{x \in [v_{\min}, v_{\max}]} A(x) \neq Z(x)$.

If $A(v_1) < Z(v_1)$, then the adversary stops at value $v_1$. Then, the algorithm's profit is less than $\text{ALG}(v_1)$. This is because what the algorithm accepts is different from $\text{ALG}(v_1)$ only in that it accepts more items of value $v_1$ in the second stage. As $\text{ALG}(v_1) = \alpha^* v_1$ (from Eqn. (5)) and the optimum is $v_1$, the algorithm has a competitive less than $\alpha^*$.

In the other case that $A(v_1) > Z(v_1)$, consider $v_2 := \arg\min_{x \in (v_1, v_{\max}]} A(x) = Z(x)$. Note that $v_2$ must exist as $A(v_{\max}) \leq 1$ and $Z(v_{\max}) = 1$. By definition of $v_2$, we have $A(v_2) = Z(v_2)$ and $A(x) \geq Z(x)$ for all $x \leq v_2$. Using integration by parts we have:

$$\int_{x=v_{\min}}^{v_2} x \cdot a(x)\mathrm{d}x = v_2 \cdot A(v_2) - v_{\min} \cdot A(v_{\min}) - \int_{x=v_{\min}}^{v_2} A(x)\mathrm{d}x$$

$$\leq v_2 \cdot Z(v_2) - v_{\min} \cdot Z(v_{\min}) - \int_{x=v_{\min}}^{v_2} Z(x)\mathrm{d}x$$

$$= \int_{x=v_{\min}}^{v_2} x \cdot z(x)\mathrm{d}x.$$

What this means is that the algorithm has obtained less profit in the first stage as oppose to $\text{ALG}(v_2)$ using the same capacity $A(v_2)$. Thus, the algorithm's value is at most $\text{ALG}(v_2) = \alpha^* v_2$. As the optimum is $v_2$, the competitive ratio is no better than $\alpha^*$.

Using the same reasoning as we used before, we know that randomization does not help. Thus, we have shown that no randomized online algorithm has a competitive ratio better than $\alpha^*$.

# E  Two-Stage Knapsack with Predictions

## E.1  Algorithm

We now extend our algorithm for the two-stage knapsack problem to the setting where we are given frequency predictions. As before, we will focus on the fully continuous version of the problem, which is exactly C-KNAPSACK-FP except with two stages. We adopt the same notation as we used in Section 3, with minor changes. Let $\text{OPT}(\mathcal{I})$ be either the optimum solution or its profit for the two-stage knapsack problem when $\mathcal{I}$ is the given input. Define $\text{OPT}(\mathcal{I}, \kappa)$ analogously but assuming that the knapsack has a reduced capacity, $\kappa$.

Because we derive our algorithm closely following the steps we took in Sections 3 and 5.1, we will omit details that are repeated. As before, we will find a competitive ratio that is close to the best competitive ratio $\alpha^*$ to an arbitrary precision using a binary search over $[\lambda, 1]$. We use the same instance $\mathcal{M}(v)$ as defined in Section 3.

**Step 1.**  Find $v^*$ such that

$$\int_{x=v^*}^{v_{\max}} x \cdot \ell(x)\mathrm{d}x = \frac{\alpha^* - \lambda}{1 - \lambda} \cdot \text{OPT}(\mathcal{M}(v^*)).$$

Under the assumptions we made for C-KNAPSACK-FP, it can be shown that $v^*$ exists. Note that this generalizes Step 1 for the single-stage knapsack with frequency predictions in Section 3 and we can recover it when $\lambda = 0$. The difference is that in addition to the value $\int_{x=v^*}^{v_{\max}} x \cdot \ell(x)\mathrm{d}x$, the algorithm can accept $\lambda(\text{OPT}(\mathcal{M}(v^*)) - \int_{x=v^*}^{v_{\max}} x \cdot \ell(x)\mathrm{d}x)$ extra profit in the second stage. Here, we used an easy observation that $\text{OPT}(\mathcal{M}(v^*))$ includes all items of value greater than $v^*$ in $\mathcal{M}(v^*)$.

**Step 2.**  In this step we define a function $\tau : [v^*, v_{\max}] \rightarrow [0, \infty)$ to determine the additional amount of items of a particular value that our algorithm would like to accept. Let $\mathcal{R}(v)$ denote an instance consisting of $u(x) - \ell(x) - \tau(x)$ items of each value $x \in [v^*, v]$ and $u(x)$ items of each value $x \in [v_{\min}, v^*]$. Let $B(v) := \int_{x=v^*}^{v} \ell(x) + \tau(x)\mathrm{d}x$ for $v \in [v^*, v_{\max}]$; and $B(v) = 0$ for $v \in [v_{\min}, v^*]$. For each $v \in [v^*, v_{\max}]$, let

$$\int_{x=v^*}^{v_{\max}} x \cdot \ell(x)\mathrm{d}x + \int_{x=v^*}^{v} x \cdot \tau(x)\mathrm{d}x + \lambda\text{OPT}(\mathcal{R}(v), 1 - B(v)) = \alpha^*\text{OPT}(\mathcal{M}(v)). \quad (9)$$

Here, the first two terms in the left-hand-side are our algorithm's profit in the first stage and the third is that in the second stage. Taking derivatives w.r.t. $v$, we solve the equations.

**Step 3.** The previous steps are performed using the predictions $\{\ell(x); u(x)\}_{x \in [v_{\min}, v_{\max}]}$ before seeing the actual items. Then, when an item of value $v$ arrives in the first stage, the algorithm accepts it if and only if $A_1(x) < B(x)$, $\forall x \geq v$, where $A_1(x)$ denotes the total amount of items of value at most $v$ that has been accepted by the online algorithm so far (in the first stage). In the second stage, the algorithm accepts the highest-valued remaining items to the full residual capacity.

**Step 4.** As before, we can find the desired $\alpha^*$ such that $B(v_{\max}) = 1$ by binary search over $[\lambda, 1]$. A minor difference from the previous binary search is that if $B(v) \geq 1$ for some $v$ in deriving $B(\cdot)$, we stop and decrease the value of $\alpha$. This is to avoid the residual capacity, i.e., $1 - B(v)$, becoming negative in Eqn. (9). However, for a fixed value $v$, we can still view $B(v)$ as a function of $\alpha^*$ and easily verify that it is continuous and mononotically increasing in $\alpha^*$ as long as $B(v) < 1$. Thus, this modified binary search is well-defined.

### E.2 Analysis

We show that the algorithm presented in the previous section indeed has a competitive ratio of $\alpha^*$ and that it is the best possible one can hope for.

At the end of the algorithm's execution, let $y$ be the largest value $v$ such that $A_1(v) = B(v)$. This implies $A_1(v) < B(v)$ for all $v > y$. Let $H$ denote all the items of value greater than $y$. Let $G$ be a set of items consisting of arbitrary $\ell(v)$ items for each $v \in [v_{\min}, v_{\max}]$ among those that actually arrive. It is an easy observation that all items in $H$ are accepted by the algorithm in the first phase, and also by the optimum solution. Let $A_1$ and $A_2$ denote the items we accepted in the first and second phases, respectively, with the items in $H$ excluded. Recall that $A_1(v)$ denotes the total size of items of value at most $v$ accepted by our algorithm in the first stage, and it is distinguished from $A_1$. Let $O$ be the items not in $H$ that are accepted by the optimum solution. As the algorithm accepts the highest-valued remaining items in the second phase, it must be that $A_2 \subseteq O \setminus A_1$. Let $v(P)$ denote the total profit of items in $P$ and $s(P)$ the total size of items in the same set. Then, the ratio of our algorithm's profit to the optimum is the following:

$$\rho := \frac{v(H \setminus G) + v(H \cap G) + v(A_1) + \lambda \cdot v(A_2)}{v(H \setminus G) + v(H \cap G) + v(O)}.$$

For the sake of contradiction suppose $\rho < \alpha^*$. Recall that $\lambda \leq \alpha^*$. Say our algorithm accepts $a_1(v)$ and $a_2(v)$ items of value $v$ in the first and second stages, respectively, and the optimum accepts $o(v)$ items of value $v$. Let $a(v) = a_1(v) + a_2(v)$.

Our proof strategy is to make a sequence of changes to modify our algorithm's solution and/or some items' value, ensuring (i) $\rho$ never increases; (ii) the instance continues respecting the prediction; (iii) the algorithm makes the optimal choices in the second stage regarding the remaining items; and (iv) the optimum solution remains optimum. After all the changes, we will show that $\rho = \alpha^*$ to draw a contradiction.

**First Change.** We first show that we can assume wlog that $a_1(x) = b(x)$ for all $x \in [v^*, y]$ for analysis. Observe that there exists $y_1$ such that $a(x) = s(x)$ for all $x \in (y_1, v^*)$ and $a_2(x) = 0$ for all $x \in [v_{\min}, y_1)$ due to the greedy behavior of our algorithm in the second stage and the fact that all items in $H$ are accepted in the first stage. Here, we only change $a_1, a_2$, therefore, the denominator of $\rho$, which is the optimum profit, will remain unchanged. Thus, (iv) will hold. We will consider a sequence of changes that only decreases the numerator.

Assume $a_1(x) \neq b(x)$ for some $x \in [v^*, y]$ since otherwise there is nothing to prove. Let $v_3$ denote the largest $x \in (v^*, y)$ such that $a_1(x) > b(x)$; it is an easy exercise to show $v_3$ exists from the definition of $y$. Similarly, let $v_1$ denote the smallest $x \in (v^*, y)$ such that $a_1(x) < b(x)$. Note that $v_1 < v_3$ from the fact that $A_1(y) = B(y)$ and $A_1(x) \leq B(x)$ for all $x \in [v^*, y]$; and $A_1(v^*) = 0$.

We consider two cases. First consider the case that $v_3 \geq y_1$. Let $v_2 := \max\{v_1, \min(A_2)\}$, where $\min(A_2)$ denotes the min value of items in $A_2$. By definition, $v_1 \leq v_2 \leq v_3$. We make the following changes: decrease $a_1(v_3)$ by an infinitesimally small $\delta > 0$; increase $a_2(v_3)$ by $\delta$; increase

$a_1(v_1)$ by $\delta$; and decrease $a_2(v_2)$ by $\delta$. Then, the numerator, the algorithm's profit, changes by
$\delta(-v_3 + \lambda v_3 - \lambda v_2 + v_1) \leq \delta(-v_3 + \lambda v_3 - \lambda v_2 + v_2) = \delta(v_3 - v_2)(\lambda - 1) \leq 0$.

Now consider the other case that $v_3 \leq y_1$. Then, this case is easier to track the changes as we do not have to consider items in $A_2$. In this case, we decrease $a_1(v_3)$ by $\delta$ and increase $a_1(v_1)$ by $\delta$. It is trivial to see that this change decreases the numerator, i.e., our algorithm's profit.

We have shown (i). It is easy to see that (ii) and (iii) are never violated under the changes we have made. As a result of the changes, we have $a_1(x) = b(x)$ for all $x \in [v^*, y]$ and $a_2(x) = s(x) - b(x)$ for all $x \in [y_1, y]$ and $a_2(x) = 0$ for all $x \in [v_{\min}, y_1]$.

**Second Change.** In this change, we increase the value of some items in $A_2$ to make $a(x) = s(x) = u(x)$ if $a_2(x) > 0$, and possibly change $o(\cdot)$ to keep satisfying (iv). To do this, we first increase the value of items in $A_2$ maximally up to value $y$ ensuring (ii). Since $A_2 \subseteq O$, such items appear in both the numerator and denominator. But, $v(A_2)$ has a multiplier $\lambda$ in the numerator. As discussed, $\lambda < \alpha^*$. Thus, if we make these changes, we will continue to have (i). It is trivial to see that (iii) holds true. To satisfy (iv), we let the optimum solution accepts $1 - s(H)$ highest valued items from the set of items consisting of $u(x)$ items $x \in [v_{\min}, v^*]$; here, we let $s(x) = u(x)$ for all $x \in [v_{\min}, v^*]$, and this doesn't change (iii). As this increases only the denominator, we still have (i). It is an exercise to see that we made changes respecting the prediction, so we have (ii) as well.

As a result, we have $s(x) = u(x) \; \forall x \in [v_{\min}, v^*]$, $a_1(x) = b(x) \; \forall x \in [v^*, y]$, $a_2(x) = u(x) - b(x)$ $\forall x \in [y_1, y]$ and $a_2(x) = 0 \; \forall x \in [v_{\min}, y_1]$.

**Third Change.** Here we decrease the value of some items in $H \setminus G$. As both our algorithm and the optimum solution accepts such items and get the full face value from them, it will only lower $\rho$. So, we will have (i). Specifically, we let $s(x) = u(x)$ for all $x \in [v^*, y_2]$, and $\int_{v^*}^{y_2}(s(x) - \ell(x))\mathrm{d}x = s(H \setminus G)$. It is easy to check all (i)–(iv) hold true.

**Fourth Change.** Now we only change $a_1(x)$ and $a_2(x)$ for $x \in [y, y_2]$. Thus, this will only affect the numerator. After the third change $a_1(x) = u(x)$ for all $x \in [y, y_2]$. For each $x \in [y, y_2]$, we set $a_1(x) = b(x)$ and $a_2(x) = u(x) - b(x)$. Clearly this can only decrease the numerator. And all (i)–(iv) hold true.

After all the changes, the resulting instance is $\mathcal{M}(y_2)$ and the algorithm's solution has the following form:

$$a_1(x) = \begin{cases} \ell(x) & x \in [y_2, v_{\max}] \\ b(x) & x \in [v^*, y_2] \\ 0 & \text{otherwise,} \end{cases}$$

$$a_2(x) = \begin{cases} 0 & x \in [y_2, v_{\max}] \\ u(x) - b(x) & x \in [y_1, y_2] \\ 0 & \text{otherwise,} \end{cases}$$

which is exactly what our algorithm accepts for $\mathcal{M}(y_2)$. Thus, the numerator of $\rho$ is exactly the LHS in Eqn. (9). Thus, we have shown that $\rho^* = \alpha$, which is a contradiction.

### E.3 Optimality

We show that no algorithm can have a competitive ratio better than $\alpha^*$. The proof is very similar to that for the case without predictions in Section D.3. We use the same adversarial instances $\{\mathcal{M}(v)\}$ as we used to compute $b(v)$ in Section 5.1; see Section 3.1.1 for the definition.

Consider any fixed deterministic algorithm $\mathcal{A}$ since randomization doesn't help as observed before. Adopting the notation we defined in Section E.2, let $a_1(v)$ denote the amount of items of value $v$ accepted by $\mathcal{A}$ in the first stage. Define $o(v)$ analogously for the optimum solution. Let $A_1(v)$ denote the total size of items of value at most $v$ accepted by $\mathcal{A}$ in the first stage.

If $A_1(v) = B(v)$ for all $v$, then it coincides with our algorithm whose competitive ratio is $\alpha^*$. So, let $v_1 := \arg\min_{x \in [v_{\min}, v_{\max}]} A_1(x) \neq B(x)$. We consider two cases as follows.

If $A_1(v_1) < Z(v_1)$, then the adversary declares that the instance is $\mathcal{M}(v_1)$. Then, $\mathcal{A}$ gets less profit than our algorithm because it produces an identical solution as ours except that it accepts more items of value $v_1$ in the second stage (but the same amount in both stages together). Thus, $\mathcal{A}$ obtains profit less than the LHS in Eqn. (9) and the optimum is $v_1$. This implies that $\mathcal{A}$ has a competitive ratio less than $\alpha^*$.

Consider the other case, $A_1(v_1) > B(v_1)$. Let $v_2 := \arg\min_{x \in (v_1, v_{\max}]} A_1(x) = B(x)$. Note that $v_2$ must exist as $A_1(v_{\max}) \leq 1$ and $B(v_{\max}) = 1$. As before, we can show $\int_{x=v_{\min}}^{v_2} x \cdot a_1(x)\mathrm{d}x \leq \int_{x=v_{\min}}^{v_2} x \cdot b(x)\mathrm{d}x$. But, we need a slightly stronger claim here. By definition of $v_2$, we have $A_1(v_2) = B(v_2)$ and $A_1(x) \geq B(x)$ for all $x \leq v_2$. Then, we can define a one-to-one mapping $\psi$ from the items accepted by $\mathcal{A}$ to the set consisting of $b(x)$ items for $x \in [v_{\min}, v_2]$ such that item $e$ has no greater value than $\psi(e)$. This mapping can be constructed by considering items in decreasing order of their value and mapping them sequentially.

What this means is the following. For the sake of analysis, pretend that $\mathcal{A}$ is given an option immediately after the first stage to choose $\psi(e)$ over $e$ it has accepted. It is an easy exercise to see that $\mathcal{A}$ makes all the swaps it is allowed to increase its profit before the second stage starts. Thus, we have shown that $\mathcal{A}$ obtains profit no greater than our algorithm does for $\mathcal{M}(v_2)$. This implies that $\mathcal{A}$'s competitive ratio is at most $\alpha^*$.