# OpenReview forum: "Online Knapsack with Frequency Predictions"
_NeurIPS.cc/2021/Conference — NeurIPS 2021 Poster_

### Official Review · Reviewer_YbwL · 2021-07-12

**Rating:** 6
**Confidence:** 4

**Summary:**

This paper studies the well-known online linear knapsack problem (and two of its generalizations) in the learning-augmented setting where the goal is to incorporate some ML predictions of the future online input such that the algorithm is consistent when the predictions are good and also robust in case of malicious predictions. The authors show that the threshold-based algorithms (such as the optimal worst-case algorithm for this problem) could not achieve the best competitive ratio in the learning-augmented setting and they provide a novel algorithm to attain this goal. Finally, some experiments on synthetic data sets are provided to verify the superior performance of the proposed algorithm compared to the worst-case algorithm in the setting with predictions.

**Limitations And Societal Impact:**

This is a theoretical work without any potential ethical concerns. The authors have perfectly described the scope of their proposed algorithm and its limitations.

**Main Review:**

Main strengths of the paper:
- The online knapsack problem is extremely well-motivated and therefore, obtaining learning-augmented algorithms for this problem is of potential interest to a wide audience.
- Unlike many previous works in the learning-augmented setting, the techniques used to design the algorithm are quite different from those used in the optimal worst-case algorithm for the problem and so, the theoretical contributions of the work are significant.
- The experiments are well designed to verify and highlight the theoretical results of the paper and show the superiority of the performance of the proposed algorithm.

Main drawbacks of the paper:
- It is really hard to follow all the notations used in the paper and understand the arguments of the proofs. For instance, I am not sure what $B_{\alpha}(\cdot)$ is in Step 4 (Section 3) of the paper. I did not find the definition elsewhere. It would be great if all the notations are summarized in one place for future reference (currently, it is scattered throughout pages 4 and 5)
- The authors need to precisely discuss the size of the additive loss in the competitive ratio (due to discretization) in intermediate cases where $\max_i s_i$ does not approach zero. For example, if we consider the discretization mentioned in the paragraph before Section 1.1 (with $k$ discrete values), what is the exact dependence of $\max_i s_i$ on $k$ and how fast the whole additive loss approaches 0 as $k$ grows larger.
- The proposed algorithm addresses the case where predictions are all correct and the authors mention that a linear combination of this algorithm with the optimal worst-case algorithm leads to a consistent and robust learning-augmented algorithm. However, this linear combination might not be the best way to incorporate both robustness and consistency, and designing a single algorithm that incorporates both the worst-case and predictions usually leads to better results. For instance, if similar to [7], there is a parameter $\lambda \in [0,1]$ denoting our confidence in the predictions, how would the robustness and consistency bounds depend on $\lambda$?
********************************
Post-Rebuttal Update: Thanks to the authors for their detailed responses to the raised questions. While I highly value the theoretical contributions of this work, I still believe that a detailed discussion about the dependence of the additive loss in the competitive ratio (due to discretization) on $k$ and its order of magnitude for finite $k$ is necessary to complement the result of Theorem 1. The authors have described their result in their response, but my main concern was about the dependence of the term $\max_i s_i$ on $k$ (i.e., the magnitude of the additive loss for finite $k$). Also, a precise description of the results in the case of incorrect predictions is required as well. I will keep my initial score for the paper.

----Final Update-----

After discussing the raised questions with the authors, my concerns were resolved. I will increase my score to 6 for this paper.

**Time Spent Reviewing:**

2-3 hours

---

> ### Author Response · Authors · 2021-08-08
> **B_alpha, consistency/robustness**
>
> Thank you for your careful review and comments.
>
> **Regarding $B_{\alpha}(\cdot)$**
>
> $B_{\alpha}(\cdot)$ is the same as $B(\cdot)$, which is formally defined in Step 3. Recall that in Steps 1--3, we assume that $\alpha$ is fixed and thus the $B(\cdot)$ obtained in Step 3 is implicitly a function of $\alpha$. We omitted this dependence on $\alpha$ in the notation for convenience. In Step 4, we highlight this dependence in order to compute the optimal value of $\alpha$. We will clarify these in the revision.
> If we use the discretization method discussed before Section 1.1, $|V| =  \lceil \log_{1+\epsilon} (v_{\max} /v_{\min})\rceil$; we mentioned $|V| = O(v_{\max}/v_{\min})$ in the paper omitting the dependency on $\epsilon$ for convenience. Then, Theorem 1.1 states an additive loss of $v_{\max}/v_{\min} \cdot (|V|+1) \cdot \max_i s_i$. Here, other parameters $v_{\max}$ (max item value) and $v_{\min}$ (min item value) are given as inputs/predictions, and the competitive ratio depends on $\max_i s_i$, which the algorithm doesn’t need to know a priori.
>
> **Robustness, consistency, $\lambda$**
>
> The parameter $\gamma$ we use to linearly combine the optimal worst-case algorithm (ZCL) and our algorithm can be viewed as $\lambda$ in [7]. In other words, if we place more weight on our algorithm (i.e. set $\gamma$ close to 1), then the competitive ratio will be closer to the best possible competitive ratio for instances respecting the predictions; if we put more weights on ZCL, then it will be closer to the competitive ratio of ZCL.
> One can indeed design an algorithm slightly better than a simple linear combination. For example, if we try to combine our algorithm and ZCL, we can factor in the items ZCL bought, to compute budget functions $B(\cdot)$. However, this led to a minimal improvement, so we didn’t include it in our study.
> To summarize, we study a model to achieve robustness and consistency in the presence of weak form of predictions where instances will likely be consistent with the predictions.

---

> ### Author Response · Authors · 2021-09-03
> **Response to post-rebuttal update**
>
> We appreciate the post-rebuttal comments. For clarification, the term $\max_i s_i$ (i.e. maximum size of any item) does not depend on $k$ (the number of distinct values). Most prior work on online knapsack (e.g. [29, 30]) assume that item sizes are infinitesimally small, i.e. $\max_i s_i \rightarrow 0$; indeed no competitive algorithms exist for online knapsack without such an assumption (see [26, 30]). In Theorem 1, we simply make this dependence on the maximum item sizes explicit for completeness. In summary, the additive loss in the competitive ratio is due to two separate factors - (i) the discretization of item values, and (ii) the bound on item sizes. Further, under the usual assumption of infinitesimal item sizes, this additive loss tends to zero.

---

> > ### Comment · Reviewer_YbwL · 2021-09-03
> > **Response to post-rebuttal update**
> >
> > Thanks for your clarifying comments. It seems to me that as the number of discretization points $k$ increases, $\max_{i}s_i$ should decrease (as fewer continuous item values are rounded to each discretized point) and my question was whether it's possible to precisely quantify this dependence. I am assuming that in $\max_i s_i$, the maximization is over $i\in[k]$. If the index $i$ corresponds to the $i$-th online input, you are absolutely right and $\max_i s_i$ wouldn't depend on $k$.

---

> > > ### Author Response · Authors · 2021-09-03
> > > **Clarifying the definition of s_i**
> > >
> > > That's correct; the latter is the case, not the former. The index $i$ in $\max_i s_i$ in the theorem corresponds to the $i$th arriving item and is independent of $k$.
> > >
> > > We apologize for the slight abuse of notation for using $v_i$ to correspond to both the value of the $i$th arriving item and also for indexing a value in the set $V$ of distinct values.

---

### Official Review · Reviewer_PbZr · 2021-07-14

**Rating:** 8
**Confidence:** 3

**Summary:**

The paper studies the online Knapsack problem in the new framework of Learning Augmented algorithms. In this setting the goal is to use predictions to improve the performance of online algorithms. Generally, two properties are wanted: (1) robustness which is a worst case bound of the competitive ratio of the algorithm (even if the predictions are totally incorrect) (2) consistency which is the guarantee that if the prediction are correct then the algorithm should be better than what any online algorithm can do. As noted by the authors in introduction, robustness here is trivial hence the focus of the paper is to obtain the consistency property. Interestingly, they can use rather weak predictions and still manage to obtain an improved bound over what online algorithms can do (more details below).


The formal definition of the problem is the following. We have a knapsack of unit capacity and n items are revealed in an online fashion. Each item $i$ has a profit $p_i$ and a size $s_i$ (the value of an item is defined as $v_i=p_i/s_i$). The goal is to maximize the profit without exceeding the capacity of the knapsack.  Once an item arrives, the algorithm has to decide whether to take it or leave it.


In the online setting without prediction, an algorithm with competitive ratio of $1/(1+ln(v_{max}/v_{min}))$ is known (if items have infinitely small size) where $v_{max}$ is the max value and $v_{min}$ the min value. It is also known that it is not possible to do better. This result is due to Zhou, Chakrabarty, and Lukose [ZCL08].


The prediction model is the following. Before receiving the instance, the algorithm is given a lower bound $l(v)$ on the total size of items of value $v$ for all $v$. Similarly, an upper bound $u(v)$ on the total size of items of value $v$ is given. These are called frequency predictions. The main result of the author is to obtain an algorithm that is optimal on instances for which the true total volume $s(v)$ of items of size $v$ is in-between the lower bound and the upper bound (i.e. $l(v)\leq s(v)\leq u(v)$ for all $v$), that is, if the frequency predictions are correct. Precisely they design an algorithm that is $\alpha$-competitive where $\alpha$ is the best competitive ratio on all instances for which the frequency predictions $l,u$ are correct.
There is no closed form of the value of $\alpha$ which depends in general on the predictions $l$ and $u$ in some non-trivial way. However they demonstrate that for reasonable predictions, the value of $\alpha$ is much better that what the ZCL algorithm can guarantee.


The authors also generalize their results to more general problems such as generalized one-way trading and two-stage online knapsack. Finally, they also demonstrate that their algorithm for online Knapsack outperforms ZCL in practice.


**Limitations And Societal Impact:**

Yes.

**Main Review:**

Originality/Quality: I believe the results are non-trivial and very interesting. In particular, an interesting challenge in this problem (that the authors mention) is that simply applying the technique of ZCL with a “threshold algorithm” cannot work in the learning augmented setting. This means that from a technical point of view, new ideas were needed for this problem.

Clarity: I find the paper clearly written and I believe in the correctness of the results of the paper although I did not check the appendix entirely.

Significance: This paper gives interesting results on a clean and natural problem.

I don’t see any major weakness in this work however I have a few questions/comments for the authors:

(1) In other learning augmented algorithms, there is also a concept of “smoothness” that says that if the prediction is slightly incorrect then the performance of the algorithm does not degrade a lot. Here I understand the prediction is rather weak so you already have some kind of smoothness but what happens if you realize that some frequency prediction is in fact slightly incorrect? Can you still maintain a better guarantee that online algorithms without predictions?

(2) I believe that if the lower bound is equal to the upper bound (i.e. $l(v)=u(v)$ for all $v$) then $\alpha^*=1$. Is this true? If yes, then I would suggest to include a brief explanation of this as I don’t think it is immediately clear from the description of the algorithm.

Typos:

Line 110 : outperform -> outperforms

Line 170 : smallest -> largest ?

Line 176 : a profit of at $\alpha$… -> a profit of at least $\alpha$…


==== UPDATE ==== after reading the author response and discussing with other reviewers, my initial opinion remains: the result and techniques are interesting. I think this is a nice paper.


**Time Spent Reviewing:**

3

---

> ### Author Response · Authors · 2021-08-08
> **Smoothness and alpha^*=1**
>
> We appreciate your careful review and questions.  We will fix the typos you identified in the revision.  Regarding your questions,
>
> **(1) Smoothness and robustness**
>
> As discussed in Lines 86--89 of the paper, robustness is easy to obtain in our setting by linearly combining our algorithm with the robust algorithm (ZCL).
> In addition to this robustness, we can still obtain a better guarantee than the best worst-case algorithm even if the frequency predictions turn out to be slightly incorrect. Although there are multiple ways to formalize this, consider the following set up: Suppose the total size of items that actually arrive with value v is $s_v$, and it is off by (1+𝜀) factor from the prediction $[\ell_v, u_v]$. In other words, $s_v \in [\ell_v/(1+\epsilon), u_v(1+\epsilon)]$, but for simplicity, say we only underestimated, i.e., $s_v \in [\ell_v, u_v(1+\epsilon)]$. In this case, it is not difficult to verify that all claims 3, 4, 5, and all the inequalities in Section 3.1, except the last inequality, hold true. The only difference is that OPT may be allowed to collect up to $u_v(1+\epsilon)$ items of value $v \leq \tilde{v}$, which can only increase the optimum by $(1+\epsilon)$ factor. Thus, the algorithm’s competitive ratio drops by at most $(1+\epsilon)$  factor. Over-prediction is even easier to handle.
> Thank you for this excellent question. We will add these remarks in the revision.
>
> **(2) $\alpha^\* = 1$**
>
> You’re absolutely right. A short reason is because $M(v)$ is the same for all values $v$ if the lower bound and upper bound coincide. So, there is a single instance we should worry about and our algorithm can completely keep up with the optimum solution on the single instance.  Thank you for the suggestion; we will comment on this in the revision.

---

### Official Review · Reviewer_ZGPi · 2021-07-18

**Rating:** 6
**Confidence:** 4

**Summary:**

This paper considers a variant of the classical online knapsack problem where each possible value, the size of an item is contained in a known interval (called frequency prediction). The paper proposes an algorithm and shows that its competitive ratio is optimal. Furthermore, the authors show some applications of the problem and the algorithm.



**Main Review:**

Comments:
The writing is generally good. However, Summarized descriptions of the proposed algorithms are helpful (such as the algorithmic environment of latex). Also, the protocol of the problem is not clearly written, in that the information of size or profit of item is given to the player before or after the player makes a decision to accept/reject the item. This information is important but is missing. So, I do not fully understand why “predictions” of intervals of sizes are helpful.

The technical results are non-trivial and strong, in that the proposed algorithm is optimal in some sense. But, the analyses of the paper seem to follow standard approaches of the online algorithm literature (e.g., continuous relaxation) and I am not sure how significant the techniques are.

Maybe one of my concerns is if the scope of the conference matches the topic of the paper. At a glance, the paper is more suitable for algorithm-related conferences. The problem setting might be interesting from an algorithmic perspective, but I am not aware of any realistic motivations for the formulation. This could be a weakness of the paper unless the authors address motivating applications.


**Time Spent Reviewing:**

3 hours

---

> ### Author Response · Authors · 2021-08-08
> **Protocol, techniques, and scope**
>
> We appreciate your careful review and thoughtful comments.
>
> **The protocol of the problem**
>
> In the standard adversarial online setting for the knapsack problem [30], when an item $i$ arrives, it reveals its size $s_i$ and profit $p_i$ first, and then the algorithm needs to decide whether to accept the item $i$ or not, immediately and irrevocably.  The previous work gives the best competitive algorithm for any instance where every item’s value lies between $v_\min$ and $v_\max$. In this work, we assume that the algorithm has access to predictions so that before the items actually arrive, the algorithm knows that the total size of items of value $v_i$ is at least $\ell_i$ and at most $u_i$. The advantage of this model is that it allows us to design better algorithms even with these weak predictions.  We will clarify these in the revision.
> Thanks for the suggestion: we’ll include a summary of the final algorithm in an algorithmic environment for clarity.
>
> **Algorithmic techniques**
>
> Although our algorithm and analysis look natural, to our surprise, we had to use a new idea (namely, Sentinel; please see Section 3.2) to obtain optimal algorithms. Also the analysis is not straightforward in the main problem and its extension to the two-stage online knapsack. This is because of the following reason. The algorithm is designed to remain competitive against the optimum solution for an instance that is the worst for the algorithm and is simultaneously consistent with the items that have arrived so far. In the absence of any predictions, it is easy to characterize the adversarial instance we should compete against (since the future can be the worst-case). However, with predictions given, items that have already arrived restrict the items that can arrive in the future. This necessitates a different and subtle analysis. Finally, the continuous relaxation is only for convenience purposes and as we show in the Supplementary Material, it is without loss of generality. We also include an adaptation of our algorithm for the discrete version in Appendix B.
>
> **Scope of the conference**
>
> Learning-augmented algorithms lie at the intersection of algorithm design and ML, and research on this topic has regularly appeared in ML venues.  Our work demonstrates that weak predictions that can only provide a range of values for key problem parameters can also be exploited to design better algorithms. We believe that such an approach could potentially be useful for different problems in this area.
> Further, online knapsack is a fundamental primitive in most online packing problems. For instance, [29] models the electric car charging problem using a knapsack formulation; this approach is further explored in the AAAI 2021 paper “Data-driven Competitive Algorithms for Online Knapsack and Set Cover”.

---

### Official Review · Reviewer_et4L · 2021-07-20

**Rating:** 7
**Confidence:** 3

**Summary:**

This paper revisits the online knapsack problem using machine-learned predictions in order to improve on prior the competitive ratio of classical algorithms. The authors take a different approach compared to other works in the area of learning-augmented online algorithms. Specifically, instead of having the prediction be a specific value predicting the true value of some parameter, they use intervals to indicate a range of values in which each parameter is predicted to be in. In this case, the parameters predicted are the frequencies of elements of some specific value that will arrive in the future. In the analysis, the authors compute the competitive ratio of their algorithm given that all the predictions are accurate in the sense that the true parameters indeed fall into their corresponding intervals.  The authors show that the competitive ratio of their proposed algorithm is instance optimal with regard to the predictions given and that the competitive ratio improves as the predicted intervals become shorter and match the performance of the algorithm by Zhou et’al when the intervals equal the entire domain.  The predictions are used by the algorithm to prevent too many low-valued elements being selected, while higher valued elements are predicted to arrive later. The choice of this approach is supported by a hardness result simpler threshold based algorithms cannot achieve an arbitrarily good competitive ratio. Also, an application to one-way trading is shown via a reduction to the online knapsack problem and two-stage knapsack variant, in which the algorithm can select elements in a second round (with some penalty on their value), is discussed. Finally, experimental results show the improved performance of the proposed algorithm compared to the algorithm by  Zhou et’al that does not use predictions



**Limitations And Societal Impact:**

The authors as well as myself do not see any potential negative societal impact from this work.



**Main Review:**

I believe that this paper suggests a very interesting approach for the analysis of learning augmented algorithms by introducing the predictions as an interval. This could fit very well in practice with the idea of machine learning algorithms providing confidence intervals for their predictions. One might need to be careful though, since the error from different confidence intervals could accumulate. It is conceivable that this approach could lead in new results for different problems in the area. Finally, the paper is quite well written and easy to follow.

**Time Spent Reviewing:**

3

---

> ### Author Response · Authors · 2021-08-08
> **Thanks for the review**
>
> We appreciate the careful review. Indeed, extending our proposed setting to incorporate predictions with confidence intervals is an interesting future direction.

---

### Decision · Program_Chairs · 2021-09-28

**Decision:**

Accept (Poster)

**Comment:**

The reviewers all find the results in this paper to be technically strong and interesting. This paper makes a nice contribution to the area of algorithms with ML predictions. Please address reviewer YbwL's comments regarding the additive loss and the description of the results in the case of incorrect predictions in the final version.

**Consistency Experiment:**

NeurIPS has a long history of experimentation. In 2014, NeurIPS ran an experiment in which 10% of submissions were reviewed by two independent committees to quantify the randomness in the review process. This year, we repeated a variant of this experiment to see how the quality of the review process has changed over time.  This paper was part of the experiment and was therefore assigned to two committees (consisting of reviewers, an Area Chair, and a Senior Area Chair) that reached independent decisions.  If both committees made the same recommendation, this recommendation was followed. If a single committee recommended acceptance, the paper was accepted (with the exception of a few cases in which the other committee identified what we considered a fatal flaw, e.g., an error in a key result).

This copy’s committee reached the following decision: **Accept (Spotlight)**

The other committee assigned to the paper recommended **Reject**.  You can find the other set of reviews, along with any follow up discussion with the authors here:
https://openreview.net/forum?id=rMm9d_aDtOa